

# AI augmented edge and fog computing for Internet of Health Things (IoHT)

Deepika Rajagopal and Pradeep Kumar Thimma Subramanian

Computer Science and Engineering, Vellore Institute of Technology University, Chennai, Tamil Nadu, India

## ABSTRACT

Patients today seek a more advanced and personalized health-care system that keeps up with the pace of modern living. Cloud computing delivers resources over the Internet and enables the deployment of an infinite number of applications to provide services to many sectors. The primary limitation of these cloud frameworks right now is their limited scalability, which results in their inability to meet needs. An edge/fog computing environment, paired with current computing techniques, is the answer to fulfill the energy efficiency and latency requirements for the real-time collection and analysis of health data. Additionally, the Internet of Things (IoT) revolution has been essential in changing contemporary healthcare systems by integrating social, economic, and technological perspectives. This requires transitioning from unadventurous healthcare systems to more adapted healthcare systems that allow patients to be identified, managed, and evaluated more easily. These techniques allow data from many sources to be integrated to effectively assess patient health status and predict potential preventive actions. A subset of the Internet of Things, the Internet of Health Things (IoHT) enables the remote exchange of data for physical processes like patient monitoring, treatment progress, observation, and consultation. Previous surveys related to healthcare mainly focused on architecture and networking, which left untouched important aspects of smart systems like optimal computing techniques such as artificial intelligence, deep learning, advanced technologies, and services that includes 5G and unified communication as a service (UCaaS). This study aims to examine future and existing fog and edge computing architectures and methods that have been augmented with artificial intelligence (AI) for use in healthcare applications, as well as defining the demands and challenges of incorporating fog and edge computing technology in IoHT, thereby helping healthcare professionals and technicians identify the relevant technologies required based on their need for developing IoHT frameworks for remote healthcare. Among the crucial elements to take into account in an IoHT framework are efficient resource management, low latency, and strong security. This review addresses several machine learning techniques for efficient resource management in the IoT, where machine learning (ML) and AI are crucial. It has been noted how the use of modern technologies, such as narrow band-IoT (NB-IoT) for wider coverage and Blockchain technology for security, is transforming IoHT. The last part of the review focuses on the future challenges posed by advanced technologies and services. This study provides prospective research suggestions for enhancing edge and fog computing services for healthcare with modern technologies in order to give patients with an improved quality of life.

Corresponding author
Pradeep Kumar Thimma
Subramanian,
tspradeepkumar@vit.ac.in

## INTRODUCTION

In healthcare, information technology is currently being used to produce intelligent solutions that improves medical diagnosis and makes effective treatment possible. Automated medical diagnostic systems and intelligent frameworks for health monitoring offer services in a range of environments, including homes, offices, and hospitals as well as transportation support, to significantly reduce medical visit costs while improving overall patient care (*Zhang et al., 2021*). Smart Internet of Things (IoT) sensors and general healthcare applications have completely transformed the way we approach healthcare. Wearable and embedded smart IoT sensors can gather real-time information such as user behaviour, mobility, and location. This information is gathered and analysed using machine learning (ML) or deep learning (DL) techniques to track people and uncover hidden patterns in the data in order to diagnose and alert to critical circumstances. For instance, children with chronic illnesses can now be monitored by a smart device using wearable sensors and cell phones and alert caretakers when specific markers, such as body temperature or heart rate, exceed predetermined criteria (*Zhang et al., 2021*; *García-Magariño et al., 2019*).

Internet of Health Things (IoHT) for patients includes many wearable devices available on the market, such as fitness bands, smart watches, and other wireless devices (*e.g.*, blood pressure monitors, heart rate monitors, blood glucose meters, *etc.*). These intelligent devices are used to provide personalised monitoring. We may use these smart gadgets to create reminders for things like calorie counting during the day, activity checks, blood pressure variations, appointments, and so on. When it comes to IoHT for doctors, a variety of wearable gadgets and home monitoring tools assist doctors in more effectively monitoring the health of their patients. In the event of a medical emergency, the patient's information is shared with the physician and their relatives, so that the best decision can be made. The Internet of Things enables the real-time delivery of patient information to the family of the patient. Wearables and other smart wireless technology allow us to track elderly people and tiny children at any time and from any place. In the event of an emergency, information is provided in real time, allowing us to manage and prepare various preventive measures to save lives. When it comes to IoHT for hospitals, we can use sensor-based smart technology to streamline various system functions (*Ketu & Mishra, 2021*). The IoHT makes it possible to track medical equipment in real-time, including defibrillators, wheelchairs, oxygen pumps, nebulizers, and other care supplies. Also, real-time employee tracking is possible. Cloud-based frameworks, which frequently incorporate big data analytics approaches, can provide reliable and accurate results for basic IoT applications that demand quick reaction (*Ketu & Mishra, 2021*). The IoHT is made up of three primary layers: sensor layers, personal server layers, and medical server layers. The sensor layers include heterogeneous sensors like electroencephalogram (EEG),

electrocardiography (ECG), photoplethysmography (PPG), motion, temperature, and so on. Personal server layers also include on-body and off-body coordinators. These layers are fundamentally different from traditional healthcare systems, which do not provide real-time monitoring and advice (*Ketu & Mishra, 2021*).

IoT-enabled equipment has altered healthcare by providing amazing capabilities such as monitoring the patients remotely and self-monitoring. Patients can monitor their overall health, and the doctor can provide good care. Internet-connected devices can access health data from remote locations (*Khan et al., 2014*; *Anuradha et al., 2021*; *Nema et al., 2021*). Users of smart healthcare systems can access them *via* mobile devices and get the necessary information from the cloud. In the event of network outages or bandwidth delays, cloud-based architectures can have a significant negative influence which can result in medical emergencies or even fatalities for essential medical IoT-based applications that require higher precision, real-time responses, and robust behavior (*Vijayakumar et al., 2019*). Fog computing and edge computing reduce some of the disadvantages of the cloud system and increase its overall efficiency. The fog layer is a middle layer between end devices and the cloud. A small portion of the huge input data is processed and examined at the fog layer prior to being transmitted to the cloud, hence decreasing the total latency of communication. Time-critical data is sent to the fog layer for quick processing, while data that needs to be preserved permanently is sent to the cloud. Instead of replacing them, fog increases the performance of cloud systems. The fog layer of healthcare systems performs a number of activities, including temporal data granulation, phase differentiation, risk assessment, and alarm generation. Applications run close to the user and use fog to support real-time analysis and decision-making (*Sood & Mahajan, 2018*; *Vijayakumar et al., 2019*).

Another technology that is becoming increasingly popular is edge computing. It is an entirely new computing paradigm that runs computations at the network edge (*Cao et al., 2020*). Its primary goal is to move processing closer to the data source. Edge computing is the process of bringing the cloud's networking, processing, storage, and resource capabilities to the network's edge in order to meet the critical needs of the IT sector in agile interconnection, real-time business, data optimisation, application intelligence, security, and so on. To meet the network's high bandwidth and low latency requirements, it also offers intelligent services at the edge. Edge computing (EC) can therefore be used in intelligent applications for local services and small-scale intelligent analytics whereas cloud computing is better suited to centralise the processing of large amounts of data. Future developments in IoT depend heavily on cloud and edge computing (*Cao et al., 2020*).

## Motivation

Edge and fog computing have recently become crucial in IoHT. Artificial intelligence (AI), ML-based techniques and algorithms have also been used in a number of elements of smart healthcare system, including load balancing, energy efficiency, and resource management (*Kennedy & Eberhart, 1995*; *Masdari et al., 2017*; *Houssein et al., 2021*; *Karaboga & Basturk, 2007*; *Rani, Ahmed & Rastogi, 2020*; *Canali & Lancellotti, 2019*; *Knowles & Corne, 2000*; *Goudarzi et al., 2021*). For this reason, we intend to focus on topics like edge and fog computing in healthcare models, various issues when combining fog and

edge computing in IoHT, the significance of AI in IoHT, numerous challenges, and the significance of machine learning-based resource management algorithms to address these challenges. Also in a fog- and edge-based healthcare architecture, resources can be managed and apps can be deployed using AI and ML-based resource management schemes and algorithms. A comprehensive survey is required to identify the potential concerns and future challenges posed by advanced technologies which inturn improve overall healthcare.

## The intended audience

### Healthcare professionals and practitioners

IoHT offers various benefits to healthcare professionals. IoHT comprises of wearable devices, implanted medical devices, insulin pen to monitor glucose levels, hearing aids, and automated treatment devices. Some of the benefits of IoHT includes improved management of drugs, instant reporting with monitoring, improved connectivity with affordability, reduced error in diagnosis, and improved treatment outcomes. The IoHT provides practitioners with precise diagnosis and recommendations. Patients also produce the finest outcomes and share better experiences. This review gives an overview of available advancements and networking technologies in IoHT, various challenges faced, and the contribution of machine learning and artificial intelligence to raising overall efficiency of the healthcare systems.

### Technologists and researchers

This survey provides a complete analysis of AI-augmented edge and fog computing technologies, deep learning algorithms, numerous problems, and evaluation matrices, thereby influencing future research paths for individuals working in fog computing, edge computing, ML, DL, and AI-based techniques in healthcare. Healthcare systems may function better and be more efficient if the methods and algorithms from the survey were applied to the current situation. A thorough analysis of the current trends and practices will guide the researchers to overcome future challenges.

### Healthcare industry and networking experts

The IoHT has transformed the healthcare sector by facilitating data tracking and mobilizing demand trends. IoT monitoring in the healthcare sector seeks to make knowledgeable judgments and deliver timely care. IoT healthcare applications allow for real-time tracking and alerting. They can offer more accurate treatments and better patient care as a result. This work sheds light on the current improvements and advancements in healthcare due to networking and AI technologies. It gives current technological information to the healthcare industries that work on advanced IoT applications and solutions. Also guides the networking experts about the present obstacles and future requirements of IoHT.

*Interdisciplinary collaborations*

Through the identification of holes in the current body of cutting-edge research and the formulation of research questions, the survey makes it easier for networking experts and AI specialists, who particularly operate in the healthcare sector or industry to collaborate.

# SURVEY METHODOLOGY

The most relevant studies and publications were found for our review using the following methods.

## Literature search

To guarantee thorough and objective coverage of the literature, a multi-step, methodical strategy was used in the search and selection of publications. To make sure there was a wide range of material available, Google Scholar was used to aid in the initial search.

## Popular keywords

Popular terms were chosen to capture the most pertinent studies from the relevant topic on AI augmented Edge and Fog computing for IoHT. Some of the keywords were: "IoHT," "Fog computing in IoHT," "Edge computing in IoHT", "Machine learning," "Deep learning", and "ML algorithms for IoT". These keywords were chosen to capture every aspect of the subject under evaluation.

## Refinement of the search

The search was narrowed down to the research domain by rearranging and combining the terms in different ways. To guarantee uniformity in comprehension and elucidation, only English-language articles were taken into consideration.

## Selection of the articles

After receiving a lengthy list of publications, their abstracts were carefully examined to ascertain their applicability and value in relation to the subject of the review that we have chosen. Innovation, methodological coherence, and a noteworthy addition to the body of current knowledge were among the selection criteria. Initial search was made in Google Scholar. A list of recent journal information was collected from it and based on that further search was made in the respective journals like IEEE, Elsevier, ScienceDirect *etc*. Recent manuscripts were selected from the above mentioned journals and shortlisted based on the published year and on their relevance to our topic.

## Bias mitigation

Within the parameters of relevance and quality, the selection and search process was designed to be as inclusive as possible in order to reduce any biases. To determine the scientific validity of each article on the shortlist, its approach was thoroughly scrutinized. This was done in order to make sure that the literature review presents a fair and impartial assessment of the current research, without favoring any one methodology or point of view over another.

### Final list of articles

A final selection of article was made following this methodical procedure in order to facilitate a thorough and objective examination and analysis.

### Search process

Various article from different journals were carefully selected and filtered. Figure 1 gives an overview of the various references collected in different year spans and Fig. 2 gives an overview of different categories of references selected for the review process.

### Research questions

The Research questions we used to investigate are as follows

RQ1: What are the various ML and DL techniques used in edge-based and fog based healthcare Models?

RQ2: What are the various challenges for adapting fog and edge computing in the IoHT environment?

RQ3: How various ML-based resource management algorithms help in improving the effectiveness of different environments?

RQ4: How can high security and low delay in communication be achieved in an IoHT framework with the help of current technologies?

All these research questions have been answered in the subsequent sections with appropriate references and illustrations.

## LAYERS IN INTERNET OF HEALTH THINGS

The edge-to-cloud architecture of IoHT includes three important layers as shown in Fig. 3. Figure 3 clearly gives an overview of the components, processes, and actors involved in each layer, like the edge, fog, and cloud. The edge layer comprises edge devices like smart watches and smart wheel chairs from which data is collected and sent to the fog layer. The fog layer comprises an IoT gateway, a server, and a deep learning component for processing emergency information. Medical experts have access at all levels. In the cloud layer, bulk processing of data occurs, and the processed data is used for data analysis by experts.

### The cloud layer

Even though cloud computing provides ample resources and services to conduct intricate analysis, the closest regional cloud facility may be hundreds of kilometers away from the data collection location. The cloud is the centralized IoT deployment platform, as cloud providers offer a variety of pre-built services for IoT operations (*Zhang et al., 2021*).

### The fog layer

The fog layer helps IoT-based systems overcome latency challenges because processing must occur on the fly. Using the fog layer, alerts can be issued in real time, anomalies can be detected, and necessary actions can be taken automatically. It supports local data storage by bringing computational power nearer the edge, decreasing the total response

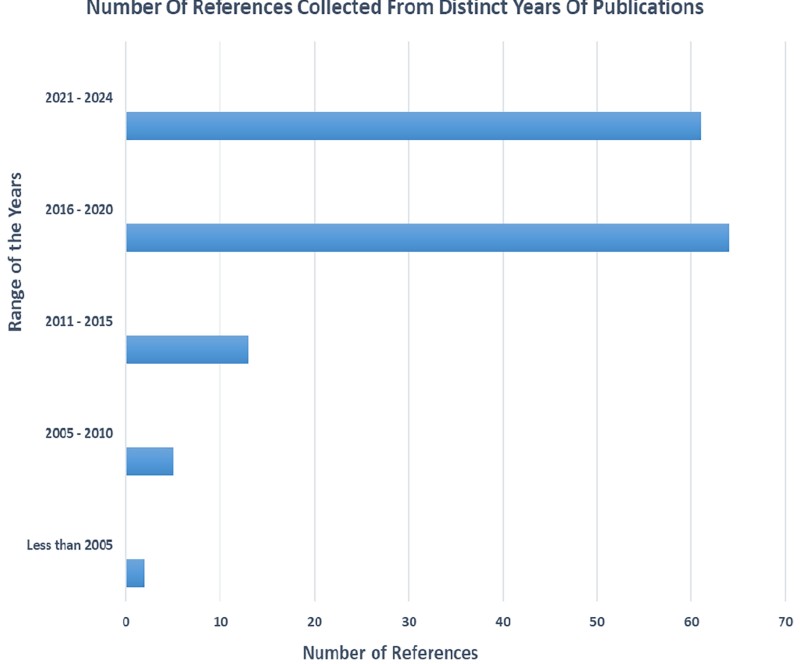

**Figure 1** The number of references collected from different ranges of years.

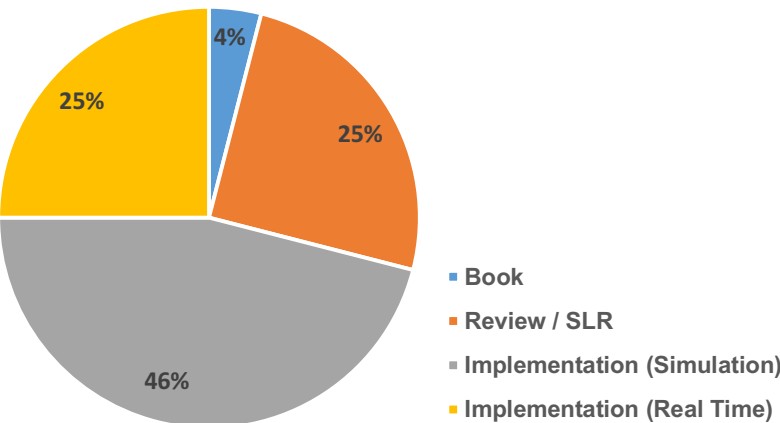

**Figure 2** Different types of references and their percentage.

time of the system (*Karakaya & Akleylek, 2021*). Fog computing settings can produce large amounts of sensor or IoT data spread over vast areas, too large to define an edge (*García-Magariño et al., 2019*). The fog layer performs operations such as preprocessing sensor data, fusion, data analysis, compression of collected data, filtering, and reducing the load on the cloud. The fog layer improves system performance, bandwidth utilisation, and system Quality of Service (QoS) (*Chudhary & Sharma, 2019*). Fog computing devices are located next to edge devices, while edge devices are positioned at the network's edge (*Omoniwa et al., 2019*; *Chakraborty et al., 2017*). Latency-aware applications can be

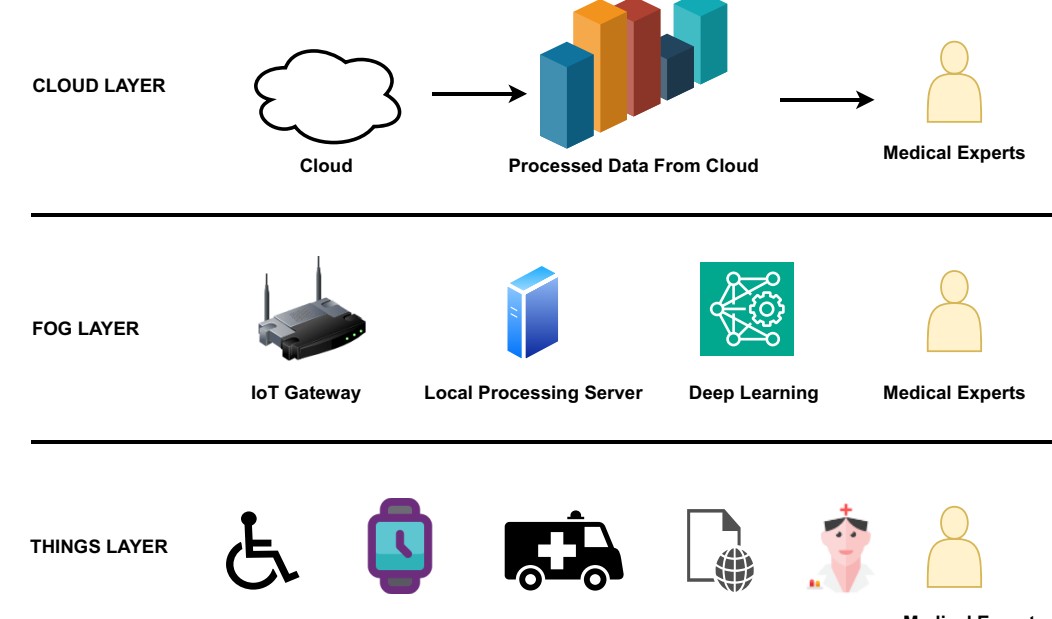

**Figure 3 The basic layers in the edge-to-cloud architecture of IoHT.**

implemented using fog computing (*Tuli et al., 2020*; *Daraghmi, Wu & Yuan, 2021*; *Naha et al., 2018*; *Bhatia & Kumari, 2022*).

Also, the fog layer modifies the task management method which is based on a priority queue and a list of fog nodes as in *Karakaya & Akleylek, (2021)*.

## The edge layer

Using processing and storage resources in close proximity to the data source is known as edge computing. Ideally, this deploys computers and storage near the data source at the network's edge. Devices like embedded automation controllers now have intelligence and processing capabilities because of edge computing. Optimising resource utilisation is just one of the many advantages that edge computing has over traditional architectures. Reducing network traffic and overcoming data bottlenecks are two other benefits of computing at the edge (*Varghese et al., 2016*).

## Protocols for communication

As shown in Fig. 4, short-range communication protocols like IEEE 802.15.1 or 802.15.4 are used for communication between a device and a fog node. An 802.11 wireless protocol connects a sensor node to additional computing devices or cloud services by utilising a sensor node, mobile computing devices, and a cloud service. Bluetooth, or IEEE 802.15.1, is used in several applications as a communication protocol between a medical device and a smartphone that provides calculations. Once the smart device has completed a brief computation, the data is transmitted to a doctor or another server *via* a mobile connection service such as 4G or 5G (*Hartmann, Hashmi & Imran, 2022*). Commercially accessible products like the Raspberry Pi, Arduino, and field programmable gate array (FPGA)

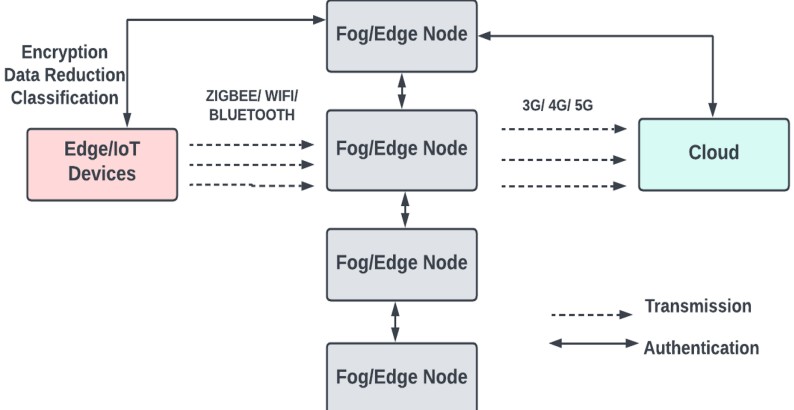

**Figure 4  The communication protocols used in the basic fog model.**

platforms are used for manufacturing edge gateways; because of their low cost and ease of use, these solutions are quite popular.

The communication protocols are mainly of two types request-reply and publish-subscribe. Message exchange in client server architecture is based on request-reply category. Representational state transfer hypertext transfer protocol (REST HTTP) and common offer acceptance protocol (CoAP) are the two main common protocols fall in this category (*Dizdarević et al., 2019*). For distributed, loosely coupled communication between data source and destination publish-subscribe protocol is preferred. Data distribution service (DDS), advanced message queuing protocol (AMQP), and message queuing telemetry transport (MQTT) are some of the examples for publish-subscribe protocols. Many works have been carried out to compare the effectiveness of these protocols in fog-to-cloud IoT architecture. Based on latency, bandwidth consumption, energy consumption and security, MQTT and hypertext transfer protocol (HTTP) are considered as most stable protocols to be used by developers for fog, cloud and IoT implementations (*Dizdarević et al., 2019*). AMQP together with REST HTTP is the main protocol used between IoT and fog layers. Also RESTful HTTP and DDS protocol proposed in *Dizdarević et al. (2019)* can be used in all layers. Mostly lightweight protocols are preferred between IoT and fog layer but that restriction is not there for communication between fog and cloud *Dizdarević et al. (2019)*. The challenge is portability and interoperability between the protocols. But in many works combination of protocols like MQTT and REST HTTP, REST HTTP and CoAP proved more efficient (*Dizdarević et al., 2019*). The usage of CoAP and HTTP protocols in IoT, fog, and cloud layers based on client server architecture is shown below in Fig. 5.

## RELATED WORKS

This section provides an overview of edge and fog computing in smart healthcare systems. Both fog and edge computing improve the performance of the cloud and increase the effectiveness of intelligent healthcare solutions. The number of IoT-based smart devices

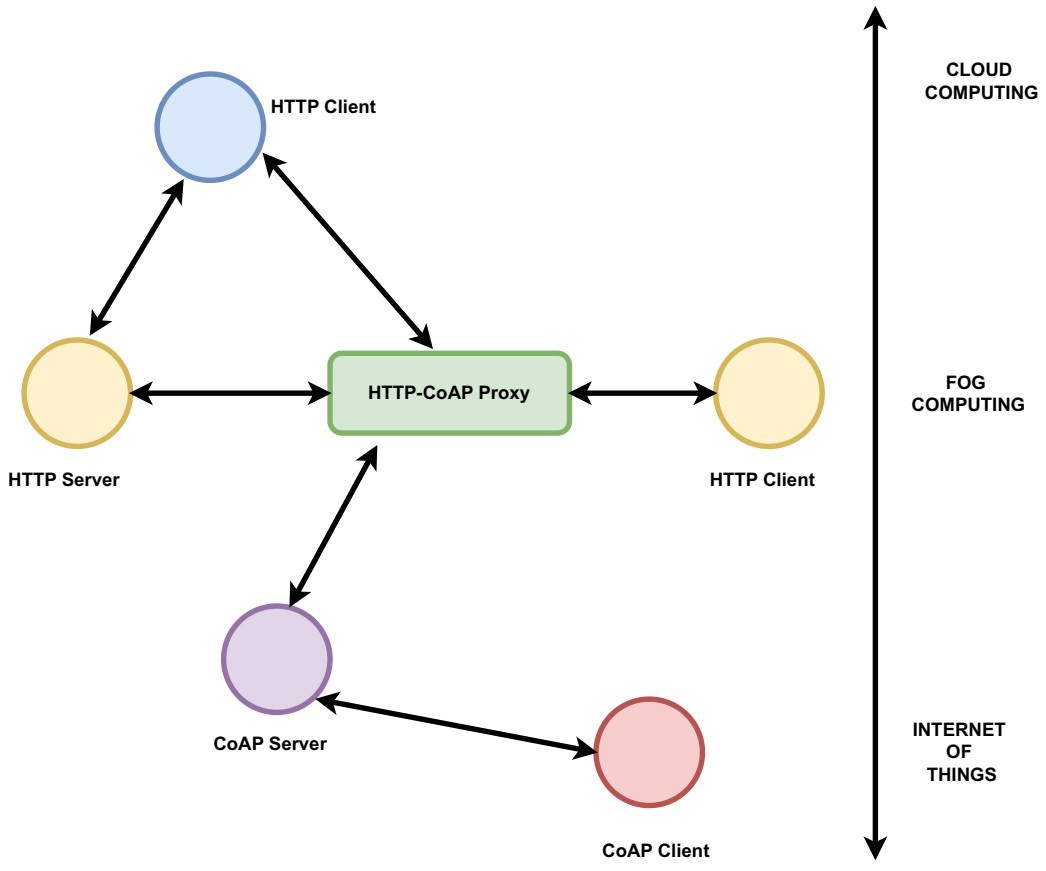

**Figure 5 HTTP-CoAP protocols connecting the IoT, fog and the cloud computing layers.**

has increased dramatically over the years. A simple bar chart shown in Fig. 6 depicts the growth of IoT devices worldwide since 2015 and their status in the future.

## Fog based healthcare model

Three critical layers make up the fog computing model for healthcare. The edge, fog, and cloud layer are the layers. End devices like sensors, medical equipment, mobile phones, *etc.*, make up the edge layer. The edge layer has billions of edge devices and all the data they produce is transferred to the fog layer to be processed. It consists of fog devices. Fog machines can perform various activities such as data separation and data analysis, and even deep learning modules can be run in the fog machines (*Di Biasi et al., 2021*) that help analyse the data and make predictions. Emergency information can be sent to the edge devices after analysing fog, as this reduces the overall response time. By using a fog layer between the edge and the cloud layer, latency is significantly reduced. Only data that needs to be stored permanently and is less sensitive is stored in the cloud. An example of fog-based diagnostics using mobile devices as an edge device is shown in Fig. 7. According to the figure, the mobile app receives the patient's data and sends it to the relevant fog node for processing. The segregation and data analysis take place there in the fog node, and the information is stored locally in the fog. Only permanently stored information is sent to the

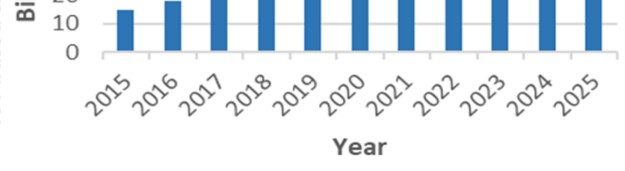

**IoT Devices Worldwide**

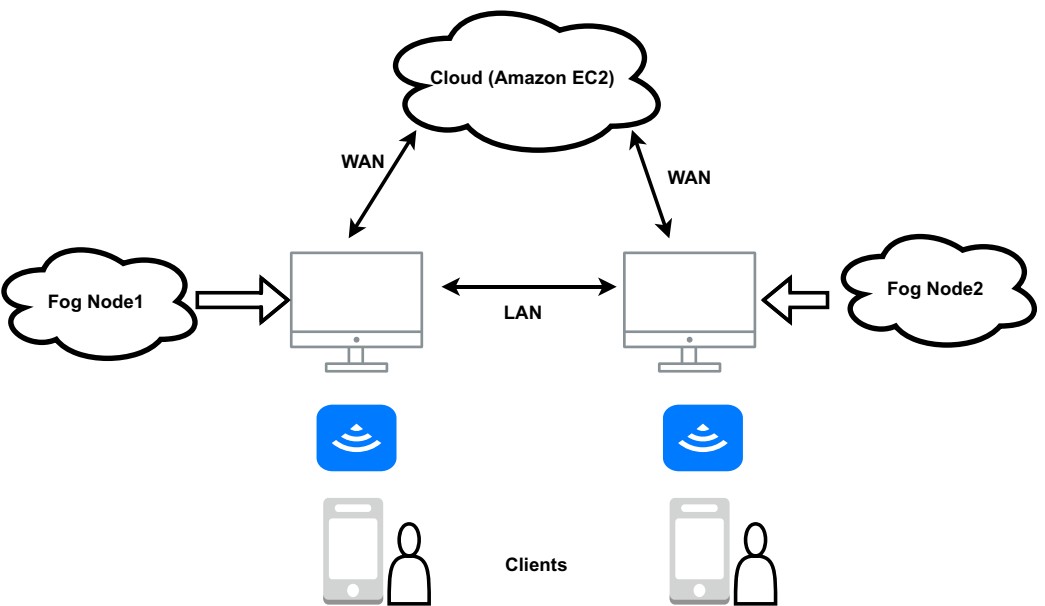

**Figure 6 The growth of IoT-based smart devices over the years.**

**Figure 7 Basic framework for fog computing based healthcare.**

cloud. The routers are connected to each other *via* Local Area Network (LAN) and to the Amazon EC2 Cloud *via* Wide Area Network (WAN). The "Wireless Access Point" (WAP) function is also integrated into the routers. This allows access to both the Amazon EC2 Cloud and the fog nodes *via* mobile and smart devices (*García-Magariño et al., 2019*; *Sood & Mahajan, 2018*; *Vijayakumar et al., 2019*; *Bhatia & Kumari, 2022*; *Chudhary & Sharma, 2019*).

## Edge based healthcare model

Wearable devices such as smartwatches, tablets, phones, and many other embedded systems perform low-level processing and act as edge devices. Edge computing moves some of the data away from the central data centre and stores it close to where the data

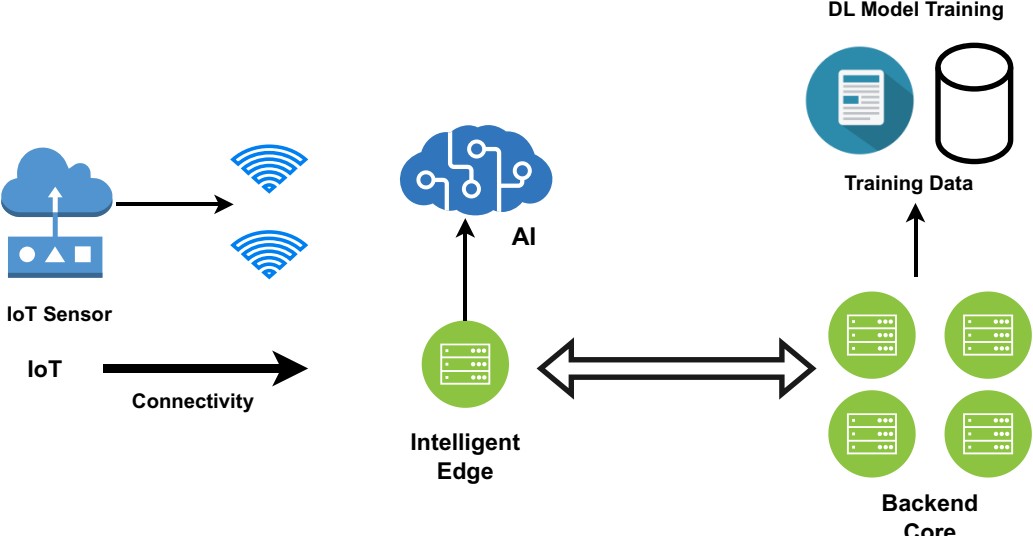

Figure 8 The edge computing framework for smart healthcare.

originated. As shown in Fig. 8, artificial intelligence tasks and data analysis techniques are performed by the edge devices, but the majority of the tedious processing and analysis is performed by the backend machine learning core, which in turn applies useful algorithms for deep analysis (*Tuli et al., 2020*; *Zhang et al., 2018*; *Abdellatif et al., 2019*; *Carvalho et al., 2021*; *Hassan, Yau & Wu, 2019*; *Ketu & Mishra, 2021*). For cloud-based computing models, data transfer speed has become a bottleneck as the amount of data created at the edge rises. The reaction time will be excessively slow if all data must be transferred to the cloud. The patient's data is received by the mobile app and routed to the relevant fog node, as depicted in the figure for processing. In such situations, the data should be processed at the network interface to reduce response time and network load (*Firouzi et al., 2023*). Data is processed near the original source in the case of edge computing. Many data mining techniques are performed in the edge layer, which speeds up execution and makes results more accurate. Local data processing takes place at the edge. Since multiple local authorities are involved, the congestion problem is reduced and reliability is increased. IBM's Watson IoT platforms have been deployed in edge computing models to analyse the analytical results based on the data generated (*Li et al., 2019*). As the edge devices can only perform fewer calculations, edge-based healthcare models are preferred when the system's computational overhead is lower and data security is of high importance. Complex calculations are carried out in the fog layer or cloud.

## General architecture for edge computing

A user device, a sensor or Internet of Things device, a computationally capable smartphone, and an edge, fog, or cloud computing node make up the general architecture of an edge computing paradigm as shown in Fig. 8. It depicts the basic architecture of edge computing. An important component of the design is the interaction between the edge and

the cloud. While the emphasis of the intervention is on speed, the benefits of cloud computing are realized in terms of long-term data (*Hartmann, Hashmi & Imran, 2022*).

Edge computing includes three layers, namely the terminal, border and cloud layers. By integrating edge devices between the cloud and endpoints, the design brings cloud services to the edge (*Cao et al., 2020*). Terminal Layer: Mobile phones, IoT devices, and several other devices are included in this layer. The gadgets in the terminal layer act as both data providers and data consumers. The top layer receives and stores all raw data from the terminal layer for processing. Boundary layer: The boundary layer is the most important component of the three-layer architecture. The edge layer is positioned between end devices and the cloud and is made up of edge nodes at the network's edge. Cloud devices, with their sophisticated computational and storage capabilities, can handle a wide range of data management, analysis, and business decision support tasks (*Cao et al., 2020*; *Abdellatif et al., 2019*; *Carvalho et al., 2021*; *Hassan, Yau & Wu, 2019*). Variation of response time based on the increasing system load in the cloud-only, edge-only, and fog-only setups is depicted below in Fig. 9. An analysis of some of the related works has been made in Table 1. The works have been analyzed based on the technologies involved, and the techniques used. The technologies include cloud computing, the Internet of Things, edge computing, and fog computing, and the techniques include simulation, alert generation (exists or not), real-time analysis.

## Overview of IoHT

### Basic stages in IoHT

IoHT for patients includes a range of wearable devices, including fitness bands, smartwatches, and other wirelessly compatible gadgets (such as heart rate monitors, blood glucose monitors, blood pressure monitors, *etc.*). These state-of-the-art devices are used to set reminders for things like daily calorie intake, activity tracking, blood pressure changes, appointments, and more. Various wearables and sensor-based gadgets are available to monitor the patient's health when discussing IoHT for hospitals. In case of an urgent medical situation, notification is sent to the doctor and the family members of the patient. IoHT also makes it simple for nurses to track medical supplies such as oxygen tanks, nebulizers, defibrillators, wheelchairs and other equipment (*Sneha & Varshney, 2007*; *Milenković, Otto & Jovanov, 2006*; *Shnayder et al., 2005*; *Ketu & Mishra, 2021*). The basic infrastructure of IoHT consists of these four steps as shown in Fig. 10. An intelligent IoHT system should be built to collect or process data at one point and generate value at the next. The first step involves deploying a heterogeneous network and devices such as cameras, sensors, displays, and actuators, which are obtained in the second step. The digitized and aggregated data are pre-processed and standardized in the third stage, which is followed by the transfer of refined data to the cloud. The final step involves conducting advanced analytics on the processed data to enable better decision-making (*Alamri et al., 2013*).

### Taxonomy of IoHT

Biomedical sensors or other sensor-based technologies gather crucial physiological data from patients, which is then utilised to identify the various solutions during the processing

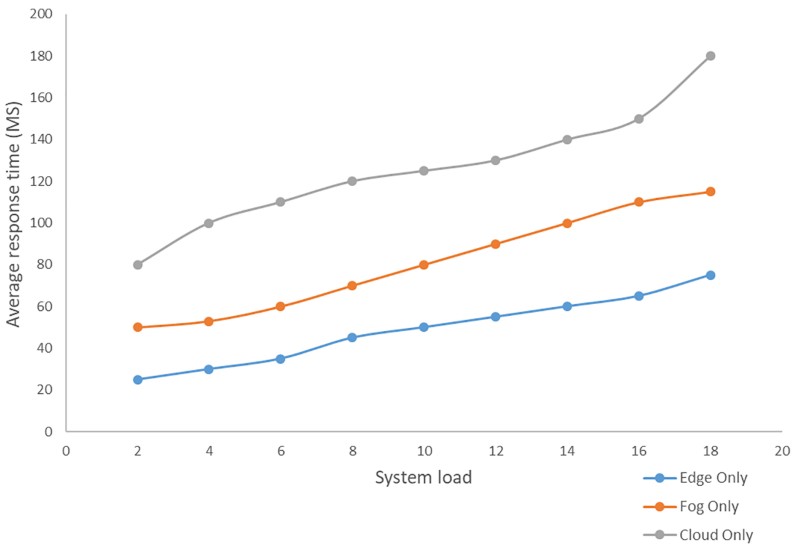

**Figure 9 Variation in average response time for different system load in cloud, fog, and edge environment.**

**Table 1 Analysis and comparison of the related works based on different technologies and techniques used.**

| Authors | Description | CC | IoT | EC | FC | SM | AG | RTA |
|---|---|---|---|---|---|---|---|---|
| *Zhang et al. (2021)* | Multi-modal deep computational model | Y | Y | Y | N | Y | N | N |
| *Anuradha et al. (2021)* | Three-tier architecture | Y | Y | N | N | Y | N | N |
| *Sood & Mahajan (2018)* | Fog cloud-based cyber-physical model | Y | Y | N | Y | Y | Y | Y |
| *Juyal, Sharma & Shukla (2021)* | Cloud based model with hierarchical CNN | Y | Y | N | N | Y | Y | N |
| *Sajjad et al. (2017)* | Multi class classification model | Y | N | N | N | Y | N | N |
| *Liu et al. (2019)* | Digital twin cloud-based healthcare model | Y | Y | N | N | Y | Y | N |
| *Li et al. (2019)* | A three layer architectural model of Edge | Y | Y | Y | N | Y | N | N |
| *Zhang et al. (2018)* | System design, Implementation, and programming interface of firework | Y | Y | Y | N | Y | Y | Y |
| *Aujla et al. (2019)* | Software defined network assisted framework for edge-cloud interplay | Y | Y | Y | N | Y | N | N |
| *Chudhary & Sharma (2019)* | Load balancing in fog-cloud environment | Y | Y | N | Y | Y | Y | N |
| *Gia et al. (2015)* | Analysis of Bio-signals for real time applications are done | Y | Y | N | Y | N | Y | Y |
| *Ahmad et al. (2016)* | Analysis of health related data using fog assisted healthcare | Y | Y | N | Y | N | N | Y |
| *Verma et al. (2022)* | Healthcare monitoring and diagnosis using deeplearning in Fog | Y | Y | N | Y | N | N | Y |
| *Silva et al. (2019)* | Medical record management using blockchain in fog computing | Y | Y | N | Y | N | N | Y |
| *Yu & Reiff-Marganiec (2021)* | IoT enabled skin disease detection using deep learning | Y | Y | N | Y | Y | Y | N |

and communication stages. Figure 11 depicts the four major levels that comprise IoHT systems: the sensor layer, the network layer, the processing layer, and the application layer.

*The sensing layer*

In IoHT systems, the sensing layer is the foremost layer. The process of acquisition involves a number of factors that includes sensor type, parameters, and method. Single-parameter and multiple-parameter collection can be done according to the needs of

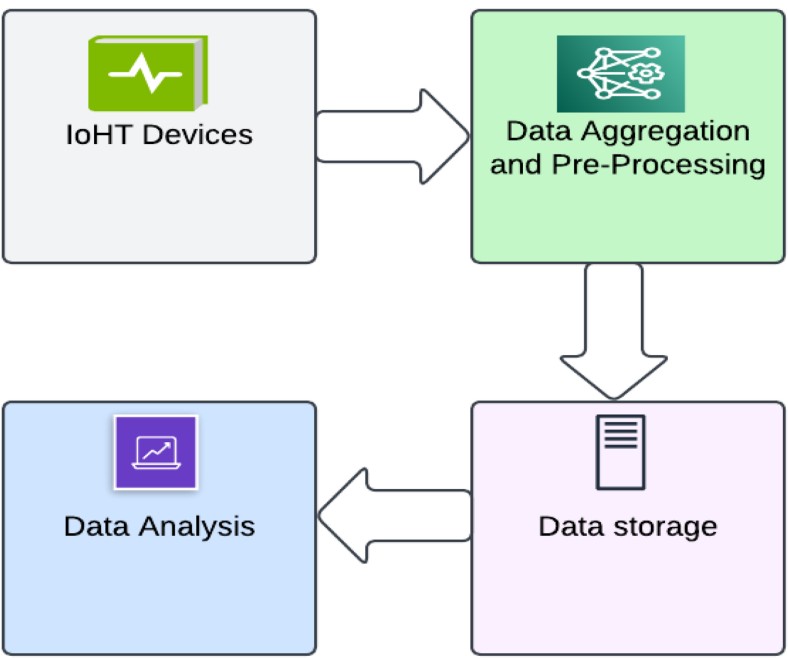

**Figure 10 Picture depicting the four basic stages in IoHT.**

intelligent healthcare systems. Perception methods include active and passive perception. Active sensing is the process of obtaining ECG data whereas passive sensing is the process of estimating heart rate using the ECG information (*Ketu & Mishra, 2021*).

*The network layer*

The second layer of IoHT systems is the network layer. It helps with both communication and the security of the system. The selection of security mechanisms and the appropriate communication protocol play an important role at this level. For heterogeneous communication, it is important to have a standard protocol for communication. Scheduling is the method we use to control time-based resource communication. This is a vital task of the IoHT system. Data management, job scheduling, and concurrent control are the three core components of the planning task. The two types of scheduling jobs are combined scheduling and publish/subscribe scheduling. While combined scheduling uses a synchronous method, publish/subscribe scheduling employs an asynchronous method. In this layer, security can be ensured at the user level and network level (*Ketu & Mishra, 2021*).

*The processing layer*

The third IoHT layer that deals with computation and data management is the processing layer. Storage type, processing medium, and data integration are all critical components of data management. The computational specification is critical in determining the fundamental technique that is integrated and used in applications based on IoHT. Some features, includes real-time remote patient monitoring, and assisting medical professionals in providing patient care. The processing layer may include deep learning or machine learning module to process the input data from sensors and aid in quick diagnosis. The

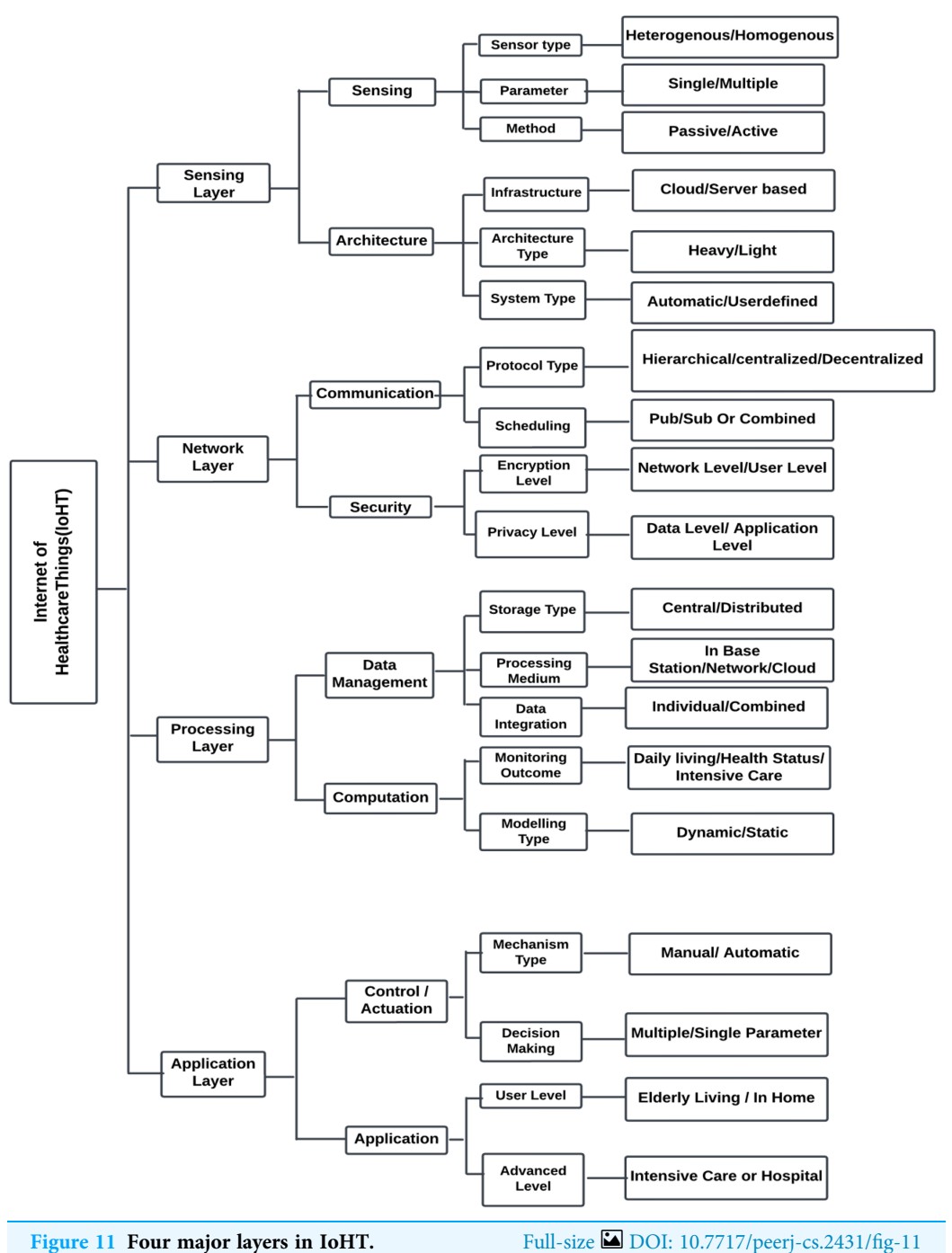

**Figure 11 Four major layers in IoHT.**

computational capacity of healthcare system has increased thanks to its real-time tracking and monitoring capability. Large-scale and sophisticated computing operations can be completed by integrating with cloud computing (*Ketu & Mishra, 2021*; *Di Biasi et al., 2021*).

*The application layer*

Next to processing layer comes the application layer. Control, actuation, and application are the important aspects of the application layer. An alarm system for patient monitoring is the best example of control/actuation in which when the input signal hits a specific threshold value the alert is activated. The alarm is triggered when the input signal reaches a particular threshold value. IoHT-based systems find use in senior care, smart homes, and hospitals. The requirements determine the application's level of complexity. It may be an application designed for beginners or for experts (*Ketu & Mishra, 2021*).

## Requirements for adapting fog computing in IoHT

The computing nodes are heterogeneous in nature and distributed in the fog layer. Safety and reliability are important factors to consider when using the fog layer (*Mahmud, Kotagiri & Buyya, 2018*). Some of the requirements for adapting fog computing in IoHT are listed below.

## Structural requirements

The fog infrastructure consists of different components with different configurations. Therefore, in fog computing, using the right strategies for information exchange between nodes and effective resource allocation is crucial (*Mahmud, Kotagiri & Buyya, 2018*; *Iftikhar et al., 2023*). The number of layers in the fog framework should be limited depending on the requirements because the number of layers affects the complexity of the network.

### Decentralised nature

The fog computing framework is decentralized, which leads to redundancy. Redundancy occurs when the same code is repeated on the edge devices (*H & Venkataraman, 2023*).

### Device heterogeneity

The end devices are of different types. This heterogeneity should be taken into account when designing the fog network at both the device level and the network level (*H & Venkataraman, 2023*). IoT applications might not be compatible with the respective architectures. For instance, certain peripherals have an Advanced RISC Machines (ARM)-based architecture and others have an Advanced Micro Device (AMD)-based design. The resource manager needs to be aware of this distinction. Another illustration is the various computing capacities offered by fog nodes. In this situation, the resource manager must be aware of the capacities of the devices in order to handle them fairly. Without it, resources would be wasted and services would be of lower quality. Since AI models are heterogeneity-agnostic and behave differently on various devices when they are implemented, controlling heterogeneity becomes even more challenging. For instance, a resource manager will take action if an edge device running AI models has accelerators like a graphics processing unit (GPU) that can offer higher precision and lower latency (*Iftikhar et al., 2023*).

## Service related requirements

Some of the fog nodes do not support the application requirements because they are not all resource-enriched. Therefore, fog should have an appropriate programming environment for creating distributed applications. Workload balancing, resource provisioning, task scheduling, and resource allocation are some of the service-related challenges. Appropriate policies need to be developed to divide tasks between IoT devices, fog nodes and the cloud. Service Level Agreements (SLAs) in fog are influenced by many factors, and it is difficult to set SLAs in the fog layer (*Mahmud, Kotagiri & Buyya, 2018*).

## Security requirements

Although edge-enabled medical equipment helps patients live better lives and opens up new revenue streams for 5G network operators and healthcare providers, there are serious privacy concerns that will only intensify as these devices are widely used. Current Health Insurance Portability and Accountability Act (HIPAA) regulations are not well-established enough to be applied to cutting-edge healthcare monitoring techniques. The healthcare provider and the network operator would both be subject to legal repercussions in the event of a data breach because many research institutes and insurance firms regard patient information as a valuable commodity (*U.S. Department of Health and Human Service, 2017*). Patient data retention laws and policies differ by nation and area, which makes matters more difficult (*Casola et al., 2016*). It is difficult to ensure security in the fog layer because there are more opportunities for attacks. The process of authentication and authorization becomes more difficult at this level. Security mechanisms can affect the QoS (*Amin & Hossain, 2021*). Areas of concern include access controls, sufficient bandwidth, user adoption, security, lack of visibility, and availability.

The integrity of the medical images is achieved with the help of a hash. Hash is more powerful than the techniques for encryption DES and AES which stands for data encryption standards and advanced encryption, respectively (*Rana, Mittal & Chawla, 2020*). Patient data should be treated with a high level of security. To further secure the images, reversible data hiding (RDH) and piecewise linear chaotic map techniques were used (*Yang et al., 2019*). The use of firewalls, virtual private networks (VPN), encryption and masking increases security (*Joshi et al., 2021*). To secure communication between sensors, a key generation scheme based on multi-biometrics in wireless body area networks (WBANs) was used (*García-Magariño et al., 2019*). The security of communication between sensors is one of the main factors to be considered in IoT-based healthcare systems. For communication between sensors, several biometric concepts were used to generate a common key. Although attribute- and identity-based encryption helps maintain data confidentiality in the cloud, the problem arises when updating the encrypted document. In addition, the transfer process from local to global leads to data leakage problems when using the cloud (*Yang, Xiong & Ren, 2020*). The question of the method by which users can securely search for specific keywords in data encrypted in the cloud has been explained in *Yang, Xiong & Ren (2020)* and *Schuiki et al. (2019)*. Many IoT security related issues in the domain of Internet of Medical Things (IoMT), Internet of

Vehicles (IoV), and intrusion prevention system (IPS) has been discussed in *Javed et al. (2023)*.

## Requirements for efficient resource management and accuracy

Resource estimation, resource discovery and resource matching are included in resource management (*Iftikhar et al., 2023*). Resource utilization, resource load, resource lifetime, response time, delay or latency, energy consumption, reliability and security are some of the metrics of fog computing (*Aslanpour, Gill & Toosi, 2020*). The difficulty of managing computation at a network's edge is a challenge faced by most applications worldwide. ML has made significant contributions to fog computing and edge processing. Fog computing would benefit greatly from the appropriate implementation of machine learning (*Abdulkareem et al., 2019*). Many factors such as application context, energy and time influence resource provisioning. Time is described in terms of calculation, communication and deadline. Computation time is the time required to complete a task. This depends on the resource configuration. Resource and power management in fog is done using task calculation time, which shows active and inactive periods of applications. Communication time defines the time required for communication in a fog environment. Communication time helps select appropriate fog nodes by defining the network context. And the deadline is the maximum service delay that a system will tolerate. Application placement plays an important role in the fog framework. To meet the desired QoS criteria, the request should be routed to the most suitable fog node based on the request and the fog resource availability (*Abdellatif et al., 2019*).

## Requirements for processing the big data

In a smart environment, IoT devices produce a lot of data. Processing this data at the edge requires real-time data analysis. Fog computing and machine learning/deep learning effectively support this process. In order to solve the problem of processing resources and reduce cloud usage costs, big data is reduced in the fog layer (*Abdulkareem et al., 2019*). Transmission latency decreases from 2.84 to 0.13 s when performing distributed computing on a smartphone. Some approaches focus on using data selection to determine whether data is transferred to a server or the cloud for further processing.

## Need for the breakdown of fog devices

This happens for many reasons such as software crashes, hardware failures and end-user activities. Multiple fog nodes are connected *via* wireless networks or Wi-Fi and the Wi-Fi connection is known to be unreliable (*H & Venkataraman, 2023*). The comparison between cloud, fog, and edge computing based on different parameters is shown in Table 2.

## Requirements to overcome the safety issues

In *Zahid et al. (2018)*, many reported data breaches are due to machine theft, manual errors, hacking, ransomware, and abuse, suggesting that basic security measures are still inadequate. Additionally, various hacking techniques such as ransomware, brute force, use of backdoors or C2, operating system commands, forced browsing, spam, and spyware/key loggers were highlighted. Deep Neural Networks, or DNNs, have been used in the fields of

**Table 2 Comparison between cloud, fog, and edge computing based on architecture, response time, latency, communication, data processing, scalability, security, and computing ability.**

|  | Cloud computing | Fog computing | Edge computing |
|---|---|---|---|
| Architecture | Centralized | Distributed | Distributed |
| Response time | High (In Minutes) | Lower (In Milliseconds) | Average (In Seconds) |
| Latency | High | Low | Low |
| Communication | From long distance | Directly with the fog node | At a verge of network |
| Data processing | Through internet | Data processed at fog node | Nearer to the source of data |
| Scalability | Easy to scale | Complex as compare to cloud | Easy as compare to fog |
| Security | Less secured | More secured | More secured than cloud |
| Computing ability | Higher | Lower | Middle-level |

image forensics, malware classification, steganalysis, and child abuse investigations in cybercrime investigations. However, no prior research has examined the robustness of DNN models in the context of digital forensics. In order to verify the security robustness of black-box DNNs, a domain-independent Adversary Testing Framework (ATF) was built in the article by *K, Grzonkowski & Lekhac (2018)*. By using ATF, a commercially available DNN service used in forensic investigations has been tested, where published methods fail in control settings. In the article *Zargari & Benford (2012)*, key aspects of digital forensics including forensic data collection, elastic, live and static forensics, evidence separation, proactive preparations, and investigations in virtualized environments in cloud forensics were discussed. Failover, instance relocation, Man in the Middle (MITM), server farming, Let's Hope for the Best (LHFTB), and address relocation techniques as well as various research challenges related to cloud forensics were discussed.

## Requirements for integration between fog nodes and the cloud

The fog framework contains both the cloud and the edge. To execute storage and processing in a distributed environment, cloud architecture and device heterogeneity should be taken into account (*H & Venkataraman, 2023*). Dynamic mapping of end-to-end fog and cloud integration should be given more importance in any fog computing architecture.

## Requirements for adapting edge computing in IoHT

There are numerous challenges associated with controlling peripheral layer devices, includes the heterogeneity of their computational and storage properties, connection, and the enormous number of devices that must be monitored at the same time. Response time, delay or latency, security, energy consumption, availability and reliability are some of metrics of edge computing (*Aslanpour, Gill & Toosi, 2020*). Nevertheless, mobility and device location are currently the biggest obstacles. Because the edge node position is malleable, the methods and protocols for controlling the node network must be developed in line with the physical edge location. Furthermore, management protocols must be designed to maintain a secure and stable system, with a particular emphasis on four factors: distinction, extensibility, isolation, and reliability (DEIR). As the number of

devices and their sensing, computing and communication capabilities continue to increase, bandwidth saturation, energy constraints and associated costs will necessitate new strategies, routines and real sustainability awareness. The future deployment of peripheral devices will depend on network virtualization technologies (*Villar-Rodriguez et al., 2023*).

## Need for collaboration of edges and heterogeneous information

Healthcare requires the collaboration of information between different stakeholders in different areas. To solve this problem, collaborative edge is used (*Abdellatif et al., 2019*). It improves overall efficiency and reduces latency and communication costs. The edge network receives input from heterogeneous end devices. To enable automatic monitoring and remote monitoring, multimodal data processing techniques must be incorporated to combine information sources. Challenges in the heterogeneous environment include transmitting informative bio signals such as EEG and electromyography (EMG), which consume more energy. The other reason is the fluctuations in the generated signals due to noise, signal offsets and other interference. To overcome this challenge, the multi-access edge computing (MEC) model was used in the S-Health system. It solves the problem of heterogeneous inputs by incorporating multimodal in-network processing methods that emphasize the temporal correlation within each modality as well as the correlation between several modalities. Using MEC, a mobile edge node (MEN) sends only limited functionality to the cloud instead of raw data. Advanced signal processing can be performed at the edge to overcome the problem of artifacts (*Abdellatif et al., 2019*).

## Privacy and security requirements

The highest potential of a healthcare system can only be achieved when the user has confidence in the security of their stored data and health-related information. The first is ownership of the data. One of the best solutions is to protect the data where it was collected and keep it close to the patient, allowing the patient to take responsibility for their own data (*Abdellatif et al., 2019*). The trade-off between security and QoS is the second challenge. Increased security levels increase computational complexity at the edge even as they increase security (*Menaka & Ponmagal, 2018*; *Omotosho, Emuoyibofarhe & Meinel, 2017*; *Masood et al., 2018*). Therefore, it is important to develop common security and QoS mechanisms that increase both the efficiency and security of the overall system (*Abdellatif et al., 2019*).

## Integration of blockchain technology in smart healthcare for security
### Blockchain technology

Blockchain technology records transactions across several computers in a decentralized, unchangeable digital ledger system that guarantees security and openness (*Ismail, Materwala & Zeadally, 2019*). A blockchain system groups data into blocks, which are then chronologically connected to form a chain. It is challenging to alter data in any block without also altering every other block in the chain since each block in the chain consists of a collection of transactions cryptographically connected to the preceding block.

### Smart contract in blockchain technology

An address-specific collection of executive codes and states is typically what makes up a smart contract. Specific condition statements in various programming languages like Solidity or Python are used to code the contracts. Once every contractor has signed a contract, a transaction is uploaded to the blockchain along with the specifications needed for smart contract functionality. The next task of the miners is to confirm the transaction and record it in an explicit block in order to generate a distinct contract address that can be used to invoke smart contracts. Following that, in order to create a unique contract address that can be used to trigger smart contracts, the miners' next responsibility is to validate the transaction and record it in an explicit block. After that, by sending a transaction to the contract, which the state variable and external reliable data sources (oracle) will validate, blockchain users can obtain the contract codes. Whenever any requirements in smart contracts are satisfied, miners will carry out and validate a matching reaction action (*Sookhak et al., 2021*). Scientists have developed nearly 40 smart contract platforms with various application and various challenges faced while using electronic health records (EHR) has been discussed in *Sookhak et al. (2021)*.

### Blockchain technology in smart healthcare

Patient data management is among the most successful applications of blockchain in the healthcare industry. Blockchain encryption methods enable safe data storage, whether it is for social security numbers, bank account information, or medical records. With the use of distributed ledger technology, patient medical records are transferred securely and are shielded from unwanted access. Additionally, blockchain guarantees data immutability (*Ismail, Materwala & Zeadally, 2019*). This ensures that patient data is accurate and authentic over time since once it is captured, it cannot be changed without the approval from the network. Furthermore, while preserving data accuracy, blockchain enables the easy exchange of patient records between various healthcare providers. The way patients are handled in the medical field has altered as a result of smart health monitoring (SHM) technologies.

## Need for integration of edge computing with 5G

Although 5G's ultra-low latency and increased bandwidth will improve edge computing, the full impact is not yet clear (*Carvalho et al., 2021*). 5G networks require significantly faster methods of data storage, decision-making, data collection strategies and remote data management, for which pre-processing needs to be improved. Because of 5G networks, the number of IoT devices will expand, which require more data processing, security, and computing resources. Additionally, managing credentials in IoT environments essentially involves huge volume of data. And the collaboration of multiple and diverse entities ensures ubiquitous computing. Innovative mobility options for communication devices could eventually become a target for network attackers. With network function virtualization (NFV) and software-defined networking (SDN), edge computing has become a key technology of the 5G era (*Carvalho et al., 2021*; *Bernardos et al., 2014*).

Additionally, the convergence of EC with 5G will enable increased response time and computing power, ignoring the physical location and resources of the IoT. The characteristics of 5G present many challenges and increase the complexity of deployment (*Carvalho et al., 2021*). According to *Hassan, Yau & Wu (2019)*, six important functions for EC in 5G networks are improving security, local data analysis, processing, decision-making, and computation, all performed locally. Edge nodes should also be able to provide some of the capabilities of the cloud.

## Mobility requirements

Mobility increases user and application flexibility but also has significant disadvantages. Mobility has a substantial impact on the loss, latency, and bandwidth of connections between edge devices and the edge network, which negatively affects the overall quality of service. User mobility can have a big impact on how many hops there are between a user and their services, especially if that movement happens at the network borders. To adapt to such network changes, edge services must be replicated quickly and dynamically. In addition to the received signal strength, the optimisation algorithms must take into account the movement direction, cost-benefit analyses of alternative networks, and service quality requirements. As the bar for resource availability, resource discovery, workload offloading, and resource provisioning is raised, the mobility of edge nodes must also be considered (*Yousefpour et al., 2019*).

## Reliability requirements

The cloud is used by edge computing devices to improve reliability. Storing data or running apps on cloud servers reduces the risk of data loss on mobile devices (*Dinh et al., 2013*). Solving issues such as platform, network, user interface, network coverage, individual device failure, and deploying edge devices without support of the cloud plays a very important role. Challenges in implementing such a system also include issues with mobility, device availability, battery limitations, connection fluctuations, and device availability. Additionally, check pointing and rescheduling methods in centralized systems do not function properly at the edge due to their mobility, heterogeneity, dynamics, and latency (*Nath et al., 2018*).

## Need for storage models

Due to problems with data ownership, data format, application programming interfaces supplied by data owners and computing platforms, controlling storage characteristics across all heterogeneous edge devices in edge cloud systems is difficult. Specific data models corresponding to the above concepts could be created for each level of the system to solve this problem. Its implementation is difficult because it must take into account a variety of factors, including automatic data consistency, device discovery, data sharing policies across nodes at different tiers, communication, and network protocols. The transversal data model must be automatic, visible to the user, and accessible at every level, regardless of the data, its location, or its load (*Wang et al., 2019*). Table 3 gives a brief comparison between our work and different surveys based on different technologies like

**Table 3 Comparison of existing surveys with our work based on different technologies, protocols, algorithms and techniques used.**

|  | IoT | Fog | Edge | Architecture | Protocol | Algorithm | Security | 5G | ML/DL | Resource management |
|---|---|---|---|---|---|---|---|---|---|---|
| *Omoniwa et al. (2019)* | Y | Y | Y | Y | Y | N | Y | N | Y | N |
| *Alamri et al. (2013)* | Y | Y | Y | Y | N | Y | Y | Y | Y | Y |
| *Iftikhar et al. (2023)* | Y | Y | Y | Y | N | Y | Y | Y | Y | N |
| *Amin & Hossain (2021)* | Y | N | Y | Y | N | N | Y | N | Y | N |
| *Abdulkareem et al. (2019)* | Y | N | Y | Y | Y | Y | Y | Y | Y | Y |
| *Ghobaei-Arani, Souri & Rahmanian (2020)* | Y | Y | N | Y | N | Y | Y | N | N | Y |
| *Kumar et al. (2022)* | Y | Y | Y | Y | N | Y | Y | N | Y | Y |
| *Dizdarević et al. (2019)* | Y | Y | Y | Y | Y | N | N | N | N | N |
| Our review | Y | Y | Y | Y | Y | Y | Y | Y | Y | Y |

IoT, fog, edge, 5G, architecture, protocols, security, ML/DL techniques and algorithms used.

## Integration of AI with 5G for effective smart healthcare and the associated challenges

AI is used in smart healthcare systems to analyse enormous volume of medical data and to deliver real-time communication and allow for the smooth integration of several procedures in healthcare (*Pradhan et al., 2023*). The introduction of 5G communication has greatly increased the opportunities for optimizing healthcare, providing low latency, fast data transfer, and the capacity to connect a wide range of systems and devices with unmatched dependability. High-speed networking, real-time data processing, and decision-making intelligence work together to provide enhanced remote monitoring, personalized treatment, better patient care, and efficient use of resources. In *Pradhan et al. (2023)*, effective integration of 5G–Multiple-input Multiple-output (MIMO) technology along with AI in an IoT environment has been discussed. Also the performance of the edge server in 5G network has been analysed based on various parameters. To guarantee the appropriate and efficient application of new technologies, however, considerable thought needs to be given to the ethical, privacy, and security considerations. In *Pradhan et al. (2023)*, some of the challenges linked to the integration of AI with 5G in IoT environment has been discussed. It includes scalability and network capacity issues, interoperability issues and lack of standards, energy efficiency issues, security and privacy problems, QoS and latency issues. Though there are few advantages, 5G technologies still faces heterogeneity and scalability issues of IoT. Also adding 5G technology to an IoT environment introduces additional security and privacy issues.

In order to provide seamless and high-performance connectivity, additional study is required to optimise QoS provisioning in 5G networks for Internet of Things applications. This includes taking into account aspects like latency reduction, traffic prioritisation, and network optimisation. In addition, research is needed to develop economic models, inorder to evaluate how cost-effective 5G-enabled Internet of Things installations are, and look at new business models that take into consideration the particular needs and

limitations of various Internet of Things applications (*Pradhan et al., 2023*). The use of new advancements like chat-bot which is user friendly is a contribution of AI (*Mah, Skalna & Muzam, 2022*).

## Edge-fog-cloud computing and blockchain technology for improving NB-IoT-based health monitoring system

The capacity and power consumption of user devices, particularly in deep coverage, can be greatly improved by integrating SHM systems with narrow band-Internet of Things (NB-IoT), which also enables covering additional categories of IoT devices and services (*Daraghmi et al., 2022*).

In order to improve NB-IoT security and authentication, *Daraghmi et al. (2022)* presents a blockchain framework that is applied to a health monitoring system that serves patients in rural and village locations. There are five primary layers in the framework: The master nodes (edges), the blockchain layer, the off-chain layer, the NB-IoT health monitoring devices, the device gateway, and so on are the first five layers. The proposal makes use of both on-chain and off-chain layers since the on-chain data contains all the necessary information to control the suggested smart contracts that are integrated into the blockchain to enhance security, authenticate users, and enable efficient data access while maintaining data integrity. Raw NB-IoT health data from verified master nodes is stored in a cloud-based database called the off-chain data.

Some of the problems faced in NB-IoT are as follows,

1) The significant latency resulting from the lack of delay-tolerant techniques in the majority of NB-IoT frameworks. Furthermore, the massive data sizes generated by the multiple terminals or the healthcare applications—like high-definition images—have an effect on the transmission time because the NB-IoT depends on the user datagram protocol (UDP) to transfer small-sized data in real time. Because the healthcare industry is so vital, it's crucial to be concerned about delays even if having a huge data set is necessary for fast throughput.

2) Security concerns since patient privacy must be preserved and sensitive patient data must be handled with care. The NB-IoT network is easily breached by hackers. Furthermore, the UDP protocol, which is used by the NB-IoT, is open to attack (*Daraghmi et al., 2022*).

To overcome the above problems, a hierarchical architecture of the network which is divided into three layers: edge, fog, and cloud computing has been proposed in *Daraghmi et al. (2022)*. The system and edge devices can communicate with each other through NB-IoT. Basic classification analysis, classification-based prioritization and an authentication procedure are all included at the edge layer. Advanced analysis, task management, and security are all included in the fog. Security and long-term data analysis are included in the cloud layer. NB-IoT has been used as the primary means of communication between edge devices and other computing layers due to its ability to cover a huge number of devices in

large areas while consuming the least amount of power. Between the edge and the fog layer is the base station.

Sorting and ranking data has been carried out to reduce base station congestion and shorten NB-IoT transmission delays. Several IoT authentication protocols has been analysed for secure NB-IoT transmissions and the most effective one has been selected. CloudSim, iFogSim and ns-3-NB-IoT are the simulators used. Authentication protocols like random MAC (RMAC), Light-Edge, and enhanced authentication and key agreement (AKA) has been tested. Access delay for with-edge and without-edge cases and execution time for no-edge, no-fog, no-edge no-fog, and the proposed architecture has been analysed. Also the authentication time taken by RMAC, Light-Edge, AKA protocol in different cases like no-edge, no-fog, no-edge no-fog, and the proposed architecture has been analysed (*Daraghmi et al., 2022*).

# CONTRIBUTION OF VARIOUS TECHNOLOGICAL ADVANCEMENTS TO INTERNET OF HEALTH THINGS

An identifiable sensor, gadget, or source that is connected to the internet and may be used in a variety of applications for automated patient monitoring to provide remote healthcare services is represented by the idea known as the Internet of Health Things (IoHT) (*Ahmed et al., 2022*). Contribution of various technological advancements to IoHT has been discussed below in this section.

## Artificial intelligence

The health sector benefits greatly from the wide variety of applications of artificial intelligence and machine learning which include prediction, classification, and analysis. For example, BlueDot Inc. (Toronto, Canada) created BlueDot: Outbreak Risk Software, which was essential in reducing the SARS-COV-2 viral outbreak (*Sheth et al., 2024*). A great deal of patient data can be analyzed using machine learning to find patterns and useful information. Thus, these cutting-edge technologies have the ability to anticipate and lessen catastrophic pandemics like COVID-19 and raise awareness in order to stop widespread breakouts (*Sheth et al., 2024*). An artificial intelligence based model was employed to monitor critical care patients during the pandemic. It is a three-stage model including input, process, and output. Within these three subparts they further divided the managerial and clinical tasks. Clinical, para-clinical, personalized medicine, and epidemiological data made up the input stage. Expert systems, deep learning, machine learning, neural networks, and artificial intelligence are all included in the process stage. Intensive care unit (ICU) decision-making, diagnosis, treatment, risk assessment, and prognosis comprised the final output stage (*Sheth et al., 2024*).

### AI on edge

The increasing demand for advanced machine learning capabilities has led to a growing importance of deploying ML systems closer to end users and data sources at the network edge. In circumstances with limited resources, edge computing provides benefits including decreased latency, more privacy and increased device mobility. When it comes to

developing and administering ML systems, the combination of edge computing and machine learning offers both new opportunities and difficulties. Many academics have conducted surveys on edge AI in the past few years as a potential replacement for IoT data processing in autonomous decision-making. An understanding of edge computing is necessary to comprehend architectures and the levels at which edge intelligence exists (*Rocha et al., 2024*). A popular method for examining real-time data close to patients is edge computing. However, edge AI—the application of AI principles in edge computing—is a recent innovation. From the initial studies in the field, integrating machine learning models into production systems has remained a barrier.

Deploying AI models from most machine learning projects in the edge is hampered by a number of challenges in this task. But when continuous software development approaches proved successful, a number of new approaches and strategies have been put out with the aim of reducing the majority of the issues that are currently present. These tactics also seek to put production advances into practice while they are being created. Machine learning operations (MLOps) is the process of implementing automation, monitoring tools, and processes at every stage of developing machine learning solutions, such as infrastructure management, testing, delivery, integration, and deployment (*Rocha et al., 2024*).

### Case study–AI in IoHT

This section presents a case study and suggests an architecture to address an important IoHT task: anomaly detection in the heart.

*End devices:* A smartwatch of the user gathers data from their PPG and ECG sensors, transmitting it to the fog/edge nodes, where a PPG2ECG model is trained. The smartwatch can use Bluetooth or Wi-Fi networks to transmit the data it has gathered to the fog/edge nodes. Subsequently, the end device applies the model and leverages it to deduce ECG information from freshly acquired real-time PPG data. The ECG data alone is insufficient to identify a cardiac abnormality. To complement the PPG2ECG model, an ECG classification model is therefore needed. The end device deploys and uses this kind of cloud-trained model to infer cardiac conditions from the ECG data that was rebuilt from the real-time PPG user data. Maintaining availability and performance of the deployed models requires ongoing maintenance and monitoring. To enhance the PPG2ECG model, the edge device periodically asks the user for fresh ECG and PPG data sets (*Rocha et al., 2024*).

*Fog/edge nodes:* In order to build a PPG2ECG model that will reconstruct an ECG signal from the PPG data, the fog/edge nodes gather PPG and ECG data from a particular user. After the model has been assessed, it is forwarded to the end device if a certain threshold metric is fulfilled. If not, the endpoint is alerted to transmit further user data for the PPG and ECG in order to train a new model with an appropriate metric value. In order to continuously obtain new data from users and enhance their PPG2ECG models, monitoring and maintenance are necessary. The devices can speak to one another using Bluetooth, Wi-Fi, or Ultra Wideband (UWB) networks (*Rocha et al., 2024*).

*Cloud:* The cloud layer is in charge of upholding the business rules as well as training, producing indications, and dumping ECG classification models that are transferred to the final layer utilizing open datasets (*Rocha et al., 2024*).

### Explainable AI

Explainable AI (XAI) techniques have been used to help clinical decision-making and guarantee trustworthy decisions while dealing with unclear information. To help with interpretability, non-ambiguous categorical qualities are converted into numerical and interpretable traits. XAI techniques that support the development of transparent and interpretable AI systems include class activation mappings (CAMs) and layer-wise relevance propagation. Numerous studies have shown how effective XAI is in medical settings, including the diagnosis of glaucoma and the identification of gestational diabetes mellitus (GDM) patients who require specialized prenatal care. In situations involving AI-assisted decision-making, explanations produced by XAI systems have raised trust and perceived fairness. Decisions made by AI systems that are biased may be unjust and untrustworthy. This emphasizes how critical XAI is to lowering the danger of bias in AI systems and guaranteeing impartial decision-making in the healthcare industry (*Albahri et al., 2023*).

### Deep learning in IoHT

*Ahmed et al. (2022)* presented an IoHT-driven, deep learning-based system for non-invasive patient discomfort detection. The new method uses an RGB camera device to measure a patient's level of discomfort instead of wearable sensors or devices. In order to recognize movement and discomfort, the system uses the YOLOv3 model to detect the patient's body and the Alphapose technique to extract numerous keypoints information of the patient's body. The identified keypoints are then transformed into six central bodily organs by applying association rules. Pairwise distance between identified keypoints is used to measure the level of pain in patients. Time frame criteria are then applied to define a threshold and examine motions to determine whether they are indicative of normal or uncomfortable conditions. Monitoring of patient's body is done in multiple frames for distinguishing abnormal conditions from normal (*Ahmed et al., 2022*).

### Advanced attention and transformer models
#### Attention models

An important development in the realm of deep learning is the attention mechanism, which was first created to improve the machine translation performance of the encoder-decoder model. Like when we focus on a single conversation in the middle of a crowded room, this mechanism works by selectively focusing on the most relevant parts of the input sequence. At its core, the neurological system in our brains that highlights relevant sounds while blocking out background distractions is similar to the attention mechanism. It enables neural networks to assign various weights to different input segments, which greatly enhances their capacity to extract critical information in the context of deep learning. In activities like natural language processing (NLP), attention is used to align pertinent portions of a source sentence during translation.

### Transformer models

Two modules make up the original transformer architecture: an encoder and a decoder. An encoder's basic unit consists of two sub-modules: a completely connected network (feed forward) and a multi-head attention (made up of several self-attention layers). Both sub-modules employ residual connections and normalization layers to stabilize the network during training. Extraction of features from input sequences is the role of the encoder. To that goal, the entire sequence is processed simultaneously and in parallel. Throughout the architecture, every token in the series follows a different path, but they are all still dependent on one another. By using this technique, the context of the entire sentence is added to each token. Depending on the workload, encoder units can be layered on top of one another. Transformers were first designed to address classic NLP problems like named entity recognition and text classification, but their exceptional sequential data handling capabilities led to the extension of the architecture to other domains where longitudinal data analysis is required, like digital health (*Siebra, Kurpicz-Briki & Wac, 2024*).

A well-known deep learning model, Transformer has found widespread use in a number of domains, including audio processing, computer vision, and NLP. Transformer was first put up as a machine translation sequence-to-sequence model. Subsequent research demonstrates that transformer-based pre-trained models (PTMs) are capable of cutting edge results across a range of activities. Transformer has consequently emerged as the preferred design in NLP, particularly for PTMs. Beyond uses pertaining to languages, Transformer has been used into computer vision (*Lin et al., 2022*). Numerous studies demonstrate that Transformer can handle enormous amounts of training data since it has a bigger capacity than convolutional neural networks (CNNs) and recurrent neural networks (RNNs). Transformer typically performs better than CNNs or RNNs when it has had enough data to train it. Also it is more flexible than CNNs and RNNs (*Lin et al., 2022*).

### Attention and transformer models for IoHT

A cardio vascular disease (CVD) prediction model for IoHT architecture has been proposed in article *Gunasekaran, Kumar & Jayashree (2024)*. With the help of adaptive residual and dilated long short term memory with attention mechanism (ARDL-AM), the pre-processed data are transmitted to the CVD prediction phase. The Enhanced Coronavirus Herd Immunity Optimizer (ECHIO) has been used for optimisation. The CVD prediction process makes advantage of the ARDL-AM results that were acquired.

In *Wang et al. (2024)*, the research presents RanMerFormer, a unique computer-aided diagnostic (CAD) approach for classifying brain tumors. The ability of CAD systems to produce precise prediction results based on medical images using cutting-edge computer vision algorithms is making it increasingly significant. The core model is a vision transformer that has been trained beforehand. After that, a merger procedure is suggested to get rid of the unnecessary tokens in the vision transformer, greatly increasing processing efficiency. Finally, the suggested RanMerFormer uses a quickly-trainable randomized vector functional-link as its head (*Wang et al., 2024*).

### Advanced vision transformer models for IoHT

A vision transformer (ViT) is a transformer designed especially for computer vision. A ViT divides an input image into several patches, serializes each patch into a vector, and then employs a single matrix multiplication to translate the image to a smaller dimension as an alternative to segmenting text into tokens. These vector embeddings are subsequently handled by a transformer encoder as token embeddings. With the demonstration of its potential for picture classification, ViT has pushed the boundaries of transformer model expertise in NLP and beyond text analysis (*Mzoughi et al., 2024*). *Mzoughi et al. (2024)* has also suggested the use of a ViT model for the multiclassification of primary brain tumors using magnetic resonance imaging (MRI) sequences. In the context of MRI-based medical image analysis, the study also offers a thorough analysis and performance comparison between the ViT model and the traditional CNN architecture.

In *Guan, Yao & Zhang (2024)*, a one-stage hybrid approach for thighbone fracture detection that combines CNN attention processes with enhanced visual transformers has been suggested. In order to maintain local continuity in X-rays, the proposed approach uses overlapping patch embedding and enhances a pyramid vision transformer architecture. A set of attention mechanisms comprising of two different types: scale-aware attention and spatial-aware attention for dynamic feature fusion across spatial and scale dimensions has been proposed. In article *Vinayahalingam et al. (2024)*, Vision transformer has been used for the multi class classification and early diagnosis of oral cancer. *Boudouh & Bouakkaz (2024)* proposes a unique method for categorizing breast calcifications in mammography with the goal of differentiating between benign and malignant occurrences. A hybrid method has been proposed to classify breast calcification.

The suggested method begins with an image pre-processing stage that uses enhancement and noise reduction filters which is followed by the hybrid classification approach. There are two branches to it: The contextual feature branch of the vision transformer (ViT++) comes first. Second, a CNN branch that uses visual feature transfer learning methods.

*Abimouloud et al. (2024)* has presented a low-weight hybrid ViT-CNN network that works directly on input patches and convolution layers to enhance feature extraction and attention layers to train patches in all networks. This method achieves accurate breast tumor classification with less parameters and a shorter training period. The suggested approach consists of dividing the input image into patches, concentrating them on the region containing malignant tumors, and supplying an input series of linear embedding of these patches. Secondly, with the least amount of adjustments feasible, a convolution layer has been applied straight to the histopathological input patches. Ultimately, patches in all transformer encoder layers has been trained to evaluate the performance of the classification of breast subtypes. In *Rodríguez, AlMarzouqi & Liatsis (2022)*, a transformer-based model optimized through significant testing, decision making and image analysis is applied for the first time in fundus multi-label illness categorization.

In *Hameed et al. (2024)*, ARiViT—a novel framework based on residual learning—has been proposed to address the challenges of brain tumor identification. ARiViT combines convolutions, adversarial learning, and vision transformers. When dealing with noisy pictures, ARiViT has remarkable resilience and adjusts effectively to different modality configurations. The classification performance of four distinct deep-learning architectures—EfficientNetB2, InceptionResNetV2, InceptionV3, and ResNet50V2—in bladder cancer detection was assessed in *Khedr et al. (2024)*. Furthermore, the potential of ViT_B32 and ViT_B16, two distinct configurations of vision transformers, was investigated for this classification task. The findings from the vision transformers were encouraging. ViT_B32 achieved an accuracy of 99.49%, while ViT_B16 achieved an accuracy of 99.23%.

## Blockchain for enhanced security in IoHT

The initial components of any IoT system centered around healthcare are sensors, RFID, and smart tags, all of which have resource constraints. Safe information transformation is crucial when these devices are integrated since they include patient data that is sensitive and might be very harmful if it ends up in the wrong hands. Blockchain-based IoT is a new technology that protects data at the physical layer by fusing the advantages of cryptography and blockchain. In *Jayaprakash & Tyagi (2021)*, elliptical curve cryptography has been used to safeguard data in blockchain network where the data is encrypted and stored.

# MACHINE LEARNING ALGORITHMS AND TECHNIQUES FOR EFFECTIVE RESOURCE MANAGEMENT IN IOT

This section explains about various existing ML algorithms for efficient application placement, Offloading, load balancing along with various nature inspired algorithms for resource management.

## Application placement algorithms

The fog service placement problem can be used to determine the best arrangement between IoT devices and fog nodes in order to utilise fog resources as efficiently as possible (*Ghobaei-Arani, Souri & Rahmanian, 2020*). *Selimi et al. (2019)* provides a micro-cloud infrastructure for community networks. Certain scenarios prioritise application placement requests to streamline application deployment to fog resources and classify the resources (*Mahmud et al., 2019*). The article explains the module mapping approach for deploying IoT application modules in the fog (*Ghobaei-Arani, Souri & Rahmanian, 2020*). To deploy the workload, the resources that are available are chosen based on the workload requirements. Fog servers must perform as many tasks as they can to use up as many resources as they can. To achieve this, one method is to use a manager or master entity. After that, the resources are distributed between fog and cloud according to the requirements of the workload (*Mijuskovic et al., 2021*). The purpose of the detection algorithms is to identify edge resources that can be used to properly distribute processing

power between the cloud and fog layers in the future (*Javaid et al., 2019*; *Taneja & Davy, 2017*).

### The round robin algorithm

The development of schedules is the foundation for the round robin (RR) algorithm's use in cloud computing. In an allocation table, the scheduler creates specific information about virtual machines (VMs). Jobs are then assigned for delivery to a group of VMs in the data centers (DCs). A virtual machine is first initialized with its current VM variable and the required work is assigned to its ID (*Javaid et al., 2019*).

### The equally spread current execution algorithm

The equally spread current execution (ESCE) algorithm uses spread spectrum technology and can simultaneously process multiple jobs running on virtual machines. In this approach, the scheduler maintains a list of all IDs of VMs with current jobs on each VM and simultaneously records the allocation table of VMs. The VM table is changed as the jobs run at each interval. The number of active tasks is zero when ESCE starts, but the scheduler selects the VM with the fewest active jobs when executing a job. The first VM is selected to process the jobs when the VMs are assigned multiple jobs with the least number. Based on the status of the VMs, different job queues are managed. It tries to distribute the work evenly among the allocated VMs after the available VMs are allocated (*Wickremasinghe, 2009*).

### The shortest job first algorithm

It executes the tasks according to their highest priority, which is determined by measuring the size of consumer requests (*Kaur & Kinger, 2014*). The distribution of jobs is based on the regions, fogs and shortest sizes of the VMs. This method is preferred over other methods due to its low latency. The waiting time for the consumer is reduced and their comfort level is not affected. First, the requests with the smallest size are fulfilled, then the appropriate VMs are allocated. The jobs are distributed among different VMs by the scheduler and the scheduling of the jobs is done by shortest job first (SJF) by allocating the minimum completion time. Additionally, it offers faster response times and greater efficiency. Following is the pseudocode for the SJF algorithm.

```
for i = 0 to i < main queue size
    if the length of task i + 1 < the length of task i then
        add task i + 1 before task i in the queue
    end if
    if main queue size = 0 then
        task i is the last in the main queue
    end if
end for
```

In the above pseudocode, the length of the tasks are compared and the task with lower execution time is given higher priority and moved forward in the queue. The average response time and average performance time of six different fog setups at six different locations using the ESCE, RR, and SJF algorithms are shown in Figs. 12 and 13, respectively

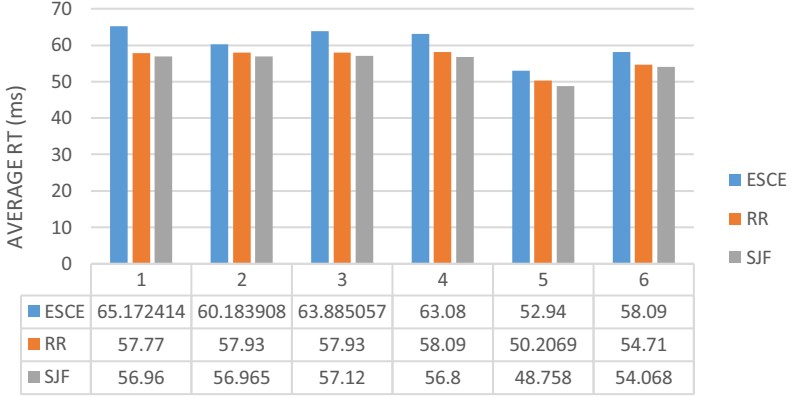

**Figure 12** Average response time in six different fog locations using ESCE, RR, and SJF algorithms.

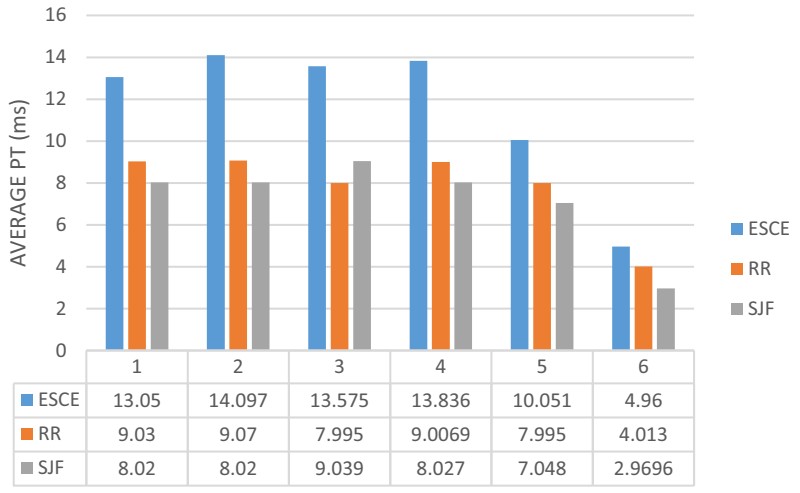

**Figure 13** Average processing time in six different fog locations using ESCE, RR, and SJF algorithms.

(*Javaid et al., 2019*). And the virtual machine costs, micro-grids and data transfer costs for the above three algorithms are shown in dollars in Fig. 14.

### Gaussian process regression for fog-cloud allocation mechanism

The first three steps of the algorithm are designed to determine the number of virtual machines that can be accommodated in the fog node, taking central processing unit (CPU) and memory into account. Step 4 invokes Gaussian process regression to predict the number of VMs that can be instantiated in the fog for future use in the next interval. If the available number of VMs that a fog node can accommodate (step 3) is greater than the number of future VMs (step 4), it is estimated that the fog node has sufficient resources to meet future demands. Otherwise, it returns null, as in the last step (Step 5) (*Da Silva & Da*

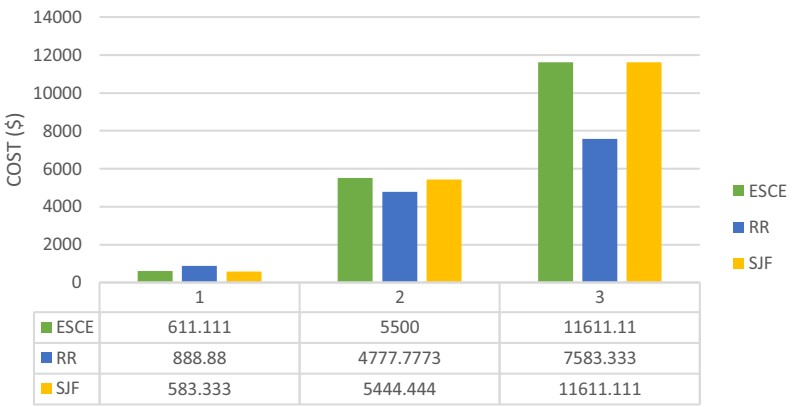

**Figure 14 Virtual machines, micro-grids, and data transfer costs for using ESCE, RR, and SJF algorithms.**

*Fonseca, 2018*). Here MIPS means million instructions per second and VM means virtual machine.

$Step$ 1: $\dfrac{availableMIPS}{mipsPerVM} \rightarrow vmsMIPS$

$Step$ 2: $\dfrac{availableRAM}{ramPerVM} \rightarrow vmsRAM$

$Step$ 3: $Min(vmsMIPS, vmsRAM) \rightarrow numberVMs$

$Step$ 4: $GaussianProcess(H) \rightarrow futureVMs$

$Step$ 5: Return $\max(0, \text{futureVMs} - \text{numberVMs})$

## Offloading algorithms

Offloading focuses primarily on resource provisioning rather than computation. It serves to minimise the overall costs. Latency is caused by determining data storage and reducing transmission costs between the cloud and IoT devices (*Wang et al., 2019*).

### Remote sync differential algorithm

The development of schedules is the foundation for the RR algorithm's use in cloud computing. In an allocation table, the scheduler creates specific information about VMs. Jobs are then assigned for delivery to a group of VMs in the data centers. A virtual machine is first initialized with its current VM (*Abdulkareem et al., 2019*).

### Fog sync differential algorithm

Although the remote sync differential algorithm (RSYNC) solves some of the problems of traditional cloud storage, too many requests are sent from the edge to the cloud, resulting in the repetition of a few data, which in places a huge load on the cloud. To solve this problem, a fog layer was introduced in the fog sync differential (FSYNC) algorithm. The fog server is deployed in the form of a cache using the FSYNC algorithm. In the FSYNC algorithm, the checksum table is sent to fog from the cloud and shared with users from fog.

The FSYNC algorithm uses a threshold. With the RSYNC, repeated synchronization requests are sent to the cloud for each update, while with FSYNC, consolidated updates are sent to the cloud, reducing communication costs and time. In FSYNC, the checksum table is updated with new information and stored as a local temporary file in fog until the threshold is reached. The cloud server is only updated by the fog when the threshold is reached. Finally, the string is reconstructed in the cloud and deleted from fog storage. The maximum or minimum rate of change is the threshold. A new file consists of two elements, namely new chunks (chunks (t + 1)) and old chunks (chunks (t)) (*Wang et al., 2019*), as shown below in Eq. (1), which is followed by the calculation of change in rate as shown in Eq. (2).

$$New_{File} = chunks(t+1) + chunks(t) \tag{1}$$
$$Change_{rate} = chunks(t+1)/old_{file} \tag{2}$$

While the cloud server is updated based on the threshold, the FSYNC algorithm checks whether the change rate (CR) is within the threshold. As shown in Eq. (3), the threshold is set as an interval [min, max]. If the CR value is within this range, the fog server will keep the sync until next time. In other cases, the file is synchronised from the fog server to the cloud. For example, let's set the interval to [0.5, 1.5]. Then the threshold f is denoted as

$$f = \begin{cases} 1, & CR \geq max \\ 0, & min < CR < max \\ 1, & CR \leq min \end{cases} \tag{3}$$

The probability of a threshold being exceeded is assumed to be x. The request times after synchronizing and changing RSYNC and FSYNC n times are shown below in Eqs. (4) and (5), respectively.

$$RSYNC = n \tag{4}$$
$$FSYNC = \sum_{t=1}^{n} f(x) \tag{5}$$

From Eqs. (4) and (5), the request time of RSYNC until x > 0 is larger than that of FSYNC. The data uploaded by RSYNC to the cloud is larger than the data uploaded by FSYNC. Therefore, FSYNC is considered more efficient than other differential synchronization methods (*Wang et al., 2019*).

### Reed-Solomon fog sync differential algorithm

The Reed-Solomon (RS) code is used to increase the security of user data. The FSYNC uses the Reed-Solomon algorithm for increased security. It also makes use of the storage capability of fog servers to address encryption issues. In addition, an erasure coding variant is used, which is used in distributed storage. The goal of Reed-Solomon fog sync differential algorithm (RS-FYNC) is to correct errors caused by the repetition of data generated from the original data (*Wang et al., 2019*).

## Load-balancing algorithm and techniques

Load-balancing algorithm (LBA) is used to manage the workload distribution in the network to prevent overload, low utilisation, and congestion which in turn improves the efficiency of operations. Some of the LBA algorithms are listed below (*Xu et al., 2018*).

### Dynamic resource allocation method

The following steps are followed in the dynamic resource allocation method (DRAM).

*Fog service partitioning:* This pre-processing method divides fog services into groups according to the demand of different types of nodes (*Xu et al., 2018*). Spare space detection: It is necessary to determine the portability of the node to support the fog service. To accommodate the fog services that are part of the same collection of services, the suitable nodes for processing are selected (*Xu et al., 2018*). The node with the least excess space is chosen during allocation. Load-balance global resource allocation: To accomplish load balance, the dynamic resource allocation approach is used (*Xu et al., 2018*).

*Spare space detection:* It is necessary to determine the portability of the node to support the fog service. To accommodate the fog services that are part of the same subset of services, the appropriate processing nodes are chosen (*Xu et al., 2018*). The node with the least excess space is chosen during allocation. Load-balance global resource allocation: To accomplish load balance, the dynamic resource allocation approach is used (*Xu et al., 2018*).

### Efficient resource allocation algorithm

This algorithm is used in a fog computing environment for efficient resource allocation. In this algorithm, an intermediate fog layer is used to reduce congestion and improve resource utilization. This algorithm works in the middle layer and the requests are forwarded to the cloud only if they are not processed within the time limit (*Agarwal, Yadav & Yadav, 2016*).

### Priority-based resource allocation algorithm

On the cloud side, the resources are examined based on the user request, and then they are given to the user based on their needs. A priority level is determined by comparing the pending request of the user with all other outstanding resources. At this time, if two requests for the same request have the same priority (first come, first served (FCFS)), the resources will be distributed based on FCFS. When calculating the throughput value, the utilization of CPU and RAM is taken into account (*Pawar & Wagh, 2012*).

### Hill climbing algorithm

Hill climbing algorithm (HCLB) is a mathematical search optimisation algorithm. Accessible VMs are found using the random solution. The loop of this algorithm runs continuously until the ideal answer to a problem is found. In HCLB, the loop is extended until the closest reachable VM is identified. The best VM is then selected, and requests are sent to it for processing (*Zahid et al., 2018*).

Figure 15 gives overview of some of the existing algorithms for efficient application placement, offloading, load balancing along with various nature inspired algorithms for resource management.

### Efficient load balancing algorithm

The architecture used in the proposed efficient load balancing algorithm (ELBA) algorithm is a four-tier architecture that includes end users, unreliable fog nodes, network resources and cloud resources. Mobile phones, laptops, connected vehicles and laptops act as fog nodes. These resources at a given location also act as fog data centers that form a cluster. And these resources change dynamically based on their availability. Nodes that are reliable in layer 3 act as controller nodes. These control nodes manage the fog nodes by classifying the busy and idle nodes and thereby forwarding the task to the appropriate node with the lowest latency. Typically, the Min-Min algorithm is used for this, which allocates the job with the shortest execution time to the resource that can process it more quickly (*Manju & Sumathy, 2019*).

## Nature-inspired algorithms

The service placement problem is used to find a mapping that assigns application components of the cloud to fog nodes, thereby achieving an optimized fitness value. It is similar to matching problems. A fog node can accept a variety of application components when one or more fog nodes are assigned to an application. This assignment can be many-to many. Application limitations, fog resource limitations, and network limitations are some of the limitations. The assignment of components is referred to by mapping, which describes whether an application component is assigned to a cloud or fog node or not. And the decision variables used are binary. For large fog and edge networks that are complex, heuristic or metaheuristic techniques are used to solve the problem. Applications running on IoT devices are interdependent. Directed Asymmetric graph (DAG) is used to model IoT applications that consist of interconnected components that represent a connected graph (*Kumar et al., 2022*).

Sensors, Internet of Things devices, smartphones, and other end-user devices with limited resources offload computationally intensive activities to remote servers to complete specified tasks (*Wang et al., 2020*). In edge technologies, computing resources are often heterogeneous, distributed, and resource-constrained with a dynamic topology and limited bandwidth linking these resources. The offloading problem is difficult and non-trivial due to the nature of processing and communication resources (*Aazam, Zeadally & Harras, 2018*). Few nature-inspired techniques are available to overcome computing offloading and service placement problems.

### Swarm intelligence-based algorithms

These algorithms were designed using the collective intelligence of species including ants, birds, honeybees, and insects. There are different types of swarm algorithms available but those that can be applied for computation offloading and service provisioning are discussed below.

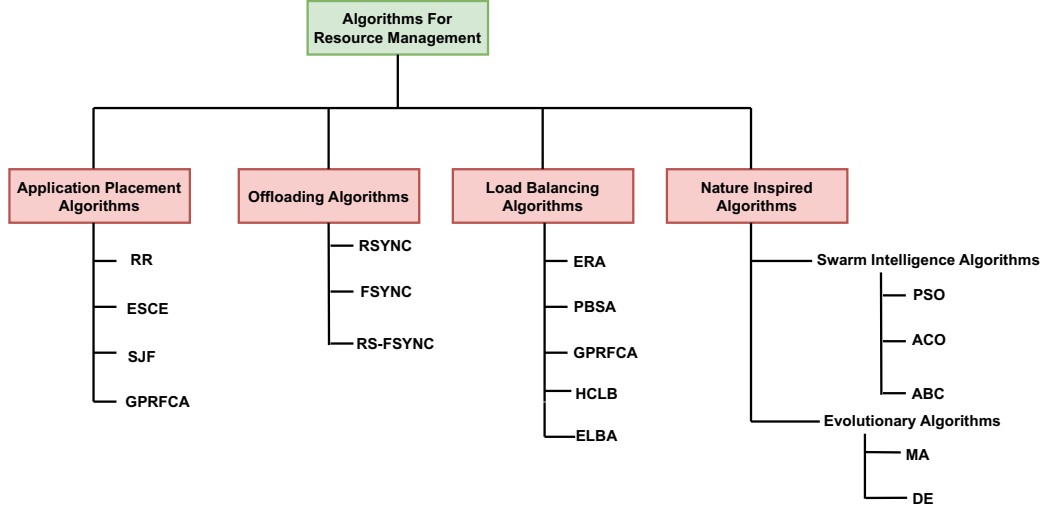

**Figure 15 List of algorithms for resource management in fog and edge environment.**

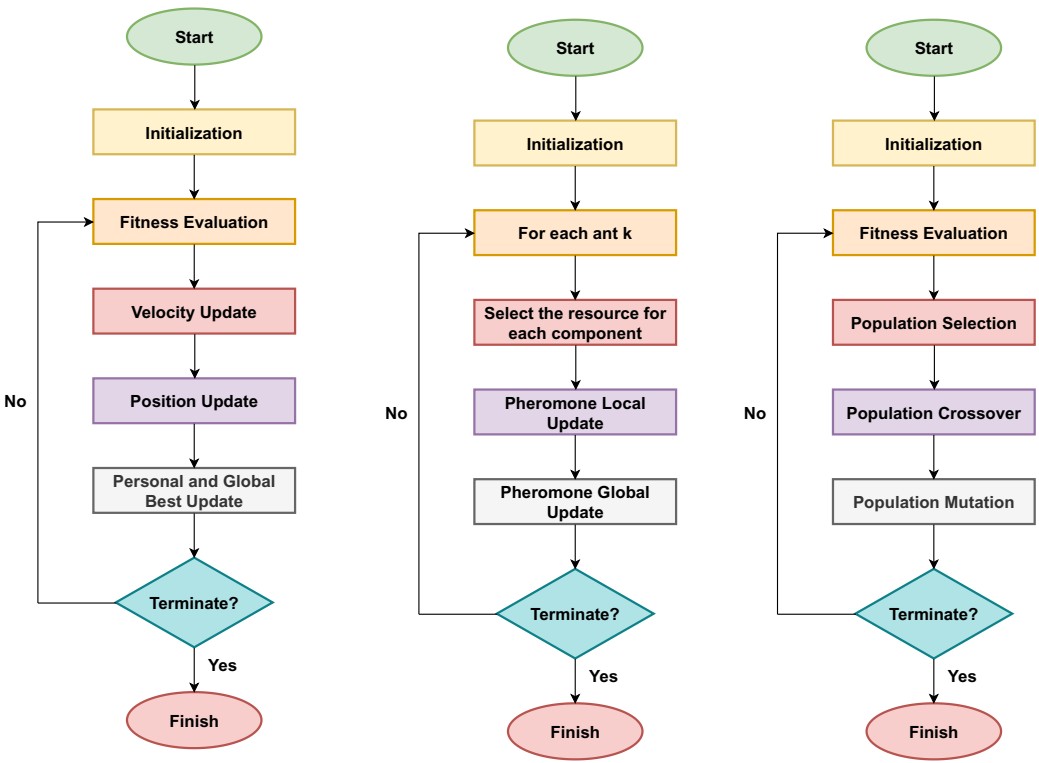

**Figure 16 Flowcharts showing different steps carried out in PSO, ACO, and GA algorithms.**

***Particle Swarm Optimization (PSO):*** The steps involved in the PSO technique are shown below in Fig. 16. A viable solution to the issue that has been optimized in the N-dimensional search space is called a particle in PSO (*Kennedy & Eberhart, 1995*).

Particles travel in accordance with their global best positions attained by all other particles as well as their local best positions. PSO is a potent optimization technique with a wide range of applications (*Masdari et al., 2017*). It does, however, have several flaws, such as sliding into local extremes and failing to guarantee accurate solutions when there are many local extremes (*Houssein et al., 2021*).

**PSO-based scheduling:** The dimension of the particle in PSO-based workflow planning is the number of tasks, and each position of the particle shows the association between the tasks and virtual machines. For example, the particle for eight tasks is shown in Figs. 17A and 17B, which describe the mapping between five VMs and eight tasks. T1, T2, …., T8 are the tasks, and VM1, VM2, …, VM5 are the virtual machines.

In *Wang & Brown (2021)*, PSO algorithm has been used in the model that detects alcoholism in patients. Using MRI of the brains, scientists have found that compared to normal, healthy individuals, brain of alcoholic patients tend to have less gray matter and white matter. On the basis of this, computer-aided diagnosis strategies for alcoholism detection have been presented in the recent past. In contrast to techniques such as support vector machines (SVM) and CNN, a unique structure for alcoholism detection has been proposed in this article. The proposed structure used a single-hidden-layer neural network trained *via* PSO as the classifier and gray level co-occurrence matrix (GLCM) as the feature extractor. The proposed structure not only showed a strong performance using experiment datasets, but it also outperformed other strategies in terms of simplicity and speed.

**Ant Colony Optimization (ACO):** Finding the best routing paths, selecting the right cluster heads, and load balancing are just some of the problems ACO is currently addressing in the IoT space (*Kumar et al., 2022*). The steps carried out in the ACO technique are shown above in Fig. 16.

### Evolutionary algorithm

Evolutionary algorithm is based on the biological processes of evolution such as reproduction, mutation, selection, and recombination.

### Genetic algorithm

This algorithm is based on the biological evolution process and is population-based (*Kumar et al., 2022*; *Rani, Ahmed & Rastogi, 2020*; *Canali & Lancellotti, 2019*). In nature, genetic algorithm (GA) exploits the idea of the survival of the fittest. Representing a chromosome, determining the fitness function, using genetic operators, and choosing the population for next generations are the four sub-processes that comprise GA. By repeatedly applying genetic operators such as crossover, mutation, and selection to the original population, a new population is eventually created. By changing the mutation and crossover probabilities, GA changes the search process. GA is easy to use, scalable, and customizable. The genetic algorithm is the best-known, most established, and most frequently used evolutionary algorithm for solving optimisation problems in various areas (*Kumar et al., 2022*). The steps carried out in GA are shown above in Fig. 16.

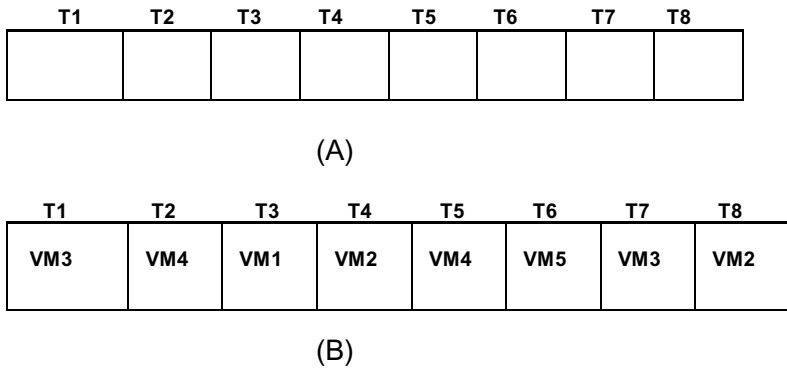

**Figure 17** (A) A particle for workflow with eight tasks. (B) The position of particle which shows the association between the tasks and virtual machines.

### Memetic algorithm

Memetic algorithm is used to solve optimisation issues and is a population-based algorithm. Meme is considered a unit of information, and the background of MA is the idea of Dawkins (*Knowles & Corne, 2000*). This method uses a mix of evolutionary algorithms and local search methods to improve the result of evolution. MA has been used to handle a variety of problems, including non-stationary function optimisation, multi-objective optimisation, training artificial neural networks, machine scheduling, and multi-class and multi-objective feature selection.

### Differential evolution

It is an evolutionary method for solving difficult optimisation problems that takes advantage of stochastic real-value coding. Differential evolution (DE) is widely used in neural network training, constrained and combinatorial optimisation, function optimisation, and clustering. Many other algorithms, coding strategies, task shifting and service placement techniques using nature-inspired algorithms have been discussed in *Kumar et al. (2022)*. Various existing works which uses ML algorithms for various resource management activities in different environment has been compared above in Table 4.

### Various application of evolutionary algorithms

*Optimisation of energy consumption*

Evolutionary algorithms has been used for variety of reasons. One such is optimization of energy consumption in IoHT with edge computing. In *Kose et al. (2024)*, an enhanced Genetic Electro-Search Optimization (GESO) method was used to optimize the energy consumption in the Internet of Health Things.

*Feature selection and diagnosis*

The process of extracting important information features from the images after pre-processing is called feature selection. Various evolutionary algorithms support this process and aid in better diagnosis. In *Zhu et al. (2023)*, evolutionary attention-based network (EDCA-Net) has been proposed. The proposed model uses two forms of evolution, such as intra- and inter-evolution. The dynamic DCA-Net, called EDCA-Net, is evaluated on four

**Table 4  List of works using ML algorithms for resource management in various environment.**

|  | Algorithms | Classification | Deployment | Environment |
|---|---|---|---|---|
| *Javaid et al. (2019)* | RR | Discovery | Simulation using Cloud Analyst | Cloud with fog |
| *Javaid et al. (2019)* | ESCE | Discovery | Simulation using Cloud Analyst | Cloud with fog |
| *Javaid et al. (2019)* | SJF | Discovery | Simulation using Cloud Analyst | Cloud with fog |
| *Da Silva & Da Fonseca (2018)* | GPRFCA | Discovery and load balancing | Simulation using (iFogSim) | Cloud with fog |
| *Abdulkareem et al. (2019)* | RSYNC | Discovery and offloading | Experiments in varying conditions | Fog |
| *Abdulkareem et al. (2019)* | FSYNC | Offloading | Experiments in varying conditions | Fog |
| *Abdulkareem et al. (2019)* | RS-FSYNC | Offloading | Experiments in varying conditions | Fog |
| *Xu et al. (2018)* | DRAM | Load-balancing | Three distinct computer methods were used for the evaluation. | Fog |
| *Agarwal, Yadav & Yadav (2016)* | ERA | Load-balancing | Simulation using Cloud Analyst | Cloud with fog |
| *Taneja & Davy (2017)* | Iterative algorithm | Placement | Three distinct topologies with different workloads were used for evaluation | Cloud with fog |
| *Arunkumar Reddy & Venkata Krishna (2021)* | FOFSA | Load-balancing | Simulation using iFogSim | Fog |
| *Zahid et al. (2019)* | HCLB | Load-balancing | Simulation using CloudAnalyst tool | Cloud with fog |
| *Manju & Sumathy (2019)* | ELBA (min-min) | Load-balancing | Simulation using CloudAnalyst tool | Cloud with fog |
| *Téllez et al. (2018)* | Tabu Search | Load-balancing | Simulation using Cloudlet tool | Cloud with fog |
| *Jiang et al. (2018)* | ECFO | Off-loading | Three Raspberry Pi3 Devices and Cloud server | Fog with edge |

**Note:**
Comparison between different works based on the environment, algorithms used, purpose and the deployment type.

openly accessible medical datasets representing a range of illnesses. The results of the experiments showed that the EDCA-Net had strong generalizability for medical image classification, outperforming the state-of-the-art techniques on three datasets and achieving equivalent performance on the fourth dataset. While the inter-evolution enables two DCA-Net instances to share training experiences during training, the intra-evolution optimizes the weights of the DCA-Net.

In *Sun & Qourbani (2023)*, a novel hybrid strategy that makes use of data mining methods like classification and feature selection has been proposed. An evolutionary algorithm and information gain are used in the integrated filter-evolutionary search approach that is used to customize feature selection. By lowering dimensions for breast cancer categorization, the suggested feature selection approach can deliver the best characteristics. Also a neural network-based ensemble classification method has been suggested in which the parameters of the neural network are modified through an evolutionary process. In the article by *Abdollahi & Nouri-Moghaddam (2022)*, the diabetic patients were classified according to the complications they had observed using an ensemble learning technique called stacked generalization based on genetic algorithms. This study combined data mining algorithms to demonstrate how combining models might enhance the model. The maximum precision was acquired by applying the

suggested Stack Generalization algorithm in compliance with the intelligence methodologies.

In *Venkata MahaLakshmi & Rout (2024)*, a novel and innovative method for diagnosing heart disease has been suggested that combines an improved ensemble classifier with an integrated filter-evolutionary search-based feature selection (iFES-FS). By combining evolutionary gravity-search based feature selection (EGS-FS) and adaptive threshold information gain-based feature selection (aTIG-FS), the suggested feature selection approach chooses the most informative features from the original feature set. *Samraj, Ramasamy & Krishnasamy (2023)* proposes a data mining method for identifying information guidelines of breast cancer analysis forecast using a genetic algorithm with histogram equalization based picture enrichment technique. In *El-Hassani et al. (2024)*, an advanced deep 1D-convolutional neural network (1DCNN) designed for diabetes classification has been proposed. It specifically addresses issues with imbalanced datasets and missing values. The best model configuration was found through three experiments: a baseline 1DCNN, hyperparameter optimization (HO) with a GA, and a PSO comparison. Results from this study are better than those from the state-of-the-art. For diagnosing lung involvement in coronavirus patients, an ant colony optimization–rain optimization approach based on hybrid deep learning has been proposed in *Challab & Mardukhi (2023)*.

*Optimisation of model's performance*
PSO, GA, simulated annealing, ant colony optimization, harmony search, and other heuristic optimization techniques have been investigated for diagnosing breast cancer. These approaches seek to repeatedly find nearly perfect solutions investigating the space of solutions. PSO draws inspiration from group behaviour and modifies a particle population according to their knowledge to identify the ideal solution. It is able to optimize parameter tuning and feature selection in breast cancer diagnosis enhancing accuracy and performance. GA is inspired through genetic evolution and natural selection. It begins with an initial group of remedies and utilizes genetic engineering to generate novel suggestions for potential solutions. The use of a GA to enhance the performance of multilayer perceptron for breast cancer diagnosis has been proposed in *Talebzadeh et al. (2024)*.

*Image enhancement*
X-ray images are most often used in diagnosis. However, the drawback is the presence of noise in the images. To overcome the problem, a novel unsupervised evolutionary method image quality assessment-based evolutionary network (IQAEvolNet) to enhance X-ray images of the chest has been proposed in *de Vasconcelos Filho, Cortez & de Albuquerque (2024)*.

## CURRENT CHALLENGES AND FUTURE WORK

To assist healthcare systems based on edge computing and successfully serve the community, a number of research challenges must be overcome.

## Edge mining for big data management in healthcare

Real-time data collection coupled with a large-scale healthcare system ensures that there is a lot of data to analyse and secure. Processing sensory data close to or at the point of perception in order to transform it from a raw signal into contextually relevant information is known as edge mining, or data mining at the edge. Edge mining techniques that reduce the volume of data transported to cloud considerably solve this problem partially; however, more reduction is vital for long-term and continuous data acquired by medical sensors. Often, rather than having to be reduced, this data can be examined in bulk. It can even be up to exobytes in size. This requires the development of new data-feature-based analysis methods (*Hartmann, Hashmi & Imran, 2022*).

## Integration of 5G for mobile edge computing-enabled health care

Current network solutions have significant hurdles when it comes to huge-scale, shared, sensor-based medical monitoring and reporting. The success of numerous supporting technologies is necessary for the transfer of IT and telecom services from a central cloud platform to an edge. Using virtualization techniques such as VMs and containers is one of the essential components. Unlike VMs, container solutions like Docker provides minimum virtualization on consumer devices which helps edge computing devices (*Ismail et al., 2015*). Similar to this, NFV separates services and network operations from proprietary hardware enabling several service instances to run concurrently on a single VM and reducing overall costs. In cases of overload due to flash crowd events, the operator of a MEC-based healthcare system can use NFV to shift system processes between two edge platforms (*Bernardos et al., 2014*). Another important technology is the use of SDNs. SDNs, or software-defined networks, are a key technology. The main idea underlying SDN is to separate the control and data planes and create a logically centralised control that allows many virtual network instances to be launched and deployed to users at the edge. In existing network topologies, managing the dynamic provisioning of various services at the network edge is challenging. SDN are anticipated to play a significant role in network connectivity and administration of services across multiple MEC systems (*Bernardos et al., 2014*; *Taleb, 2014*). Additionally, network slicing allows a single network to be divided into numerous instances, each tailored to a specific application or use case (*Taleb et al., 2017*; *Afolabi et al., 2017*). Various 5G network segments, including mobile broadband, vehicle connections, and extensive IoT may exist. Due to the high capacity requirements of enhanced mobile broadband of 5G, several other related technologies that have been introduced in the Radio Access Network (RAN) would make it possible to provide more bandwidth, pipeline packet processing, fewer TTIs, and effective radio resource control (RRC). These enabling technologies include things like user-centred designs, massive MIMO (mMIMO), millimetre wave spectrum (mmWave) transmission, and others (*Hashmi et al., 2018*; *Hashmi, Zaidi & Imran, 2018*; *Cetinkaya, Hashmi & Imran, 2018*; *Bai & Heath, 2015*; *Andrews et al., 2017*; *Rappaport et al., 2013*).

## Need for integration of AI with 5G for the growing need

The next-generation wireless cellular network (5G) is anticipated to meet the needs of next-generation networks. The following are the three main issues that the networks of the previous generation failed to address. First of all, the growing use of mobile devices generates enormous amounts of data. Second, in order to support highly interactive applications that require extremely low latency and high throughput, tight QoS criteria must be met. Third, a heterogeneous environment must be supported to ensure interoperability between a variety of user devices (*e.g.*, smartphones and tablets), QoS requirements (*e.g.*, different levels of latency and throughput for multimedia applications), network types (*e.g.*, IEEE 802.11 and the Internet of Things), *etc.* (*Hassan, Yau & Wu, 2019*). 5G is able to solve the majority of these problems. 5G networks will use high-frequency millimetre waves to strengthen communications. Mobile users would be connected to a base station that is connected to the core network *via* a common 5G mobile network. With low-latency radio interfaces, communication between the end user and the base station would be possible in less than a millisecond, but latency would increase significantly if cloud-based application requests were routed from the core network to the cloud. Fog computing can help shift computing power closer to the end user which is required for 5G networks. Real-time interaction, high data rates, high availability, and local processing are four essential conditions for edge computing to be successfully designed and implemented in 5G. While the rollout of 5G enables widespread MEC-based healthcare infrastructure, in order to offer clients the most pertinent and timely services, it is essential to integrate AI.

To enable rapid diagnosis of health issues, artificial intelligence employs a range of criteria, including user mobility patterns, device utilisation patterns, critical tracking of information from patients, and pre-existing medical disorders. Recent advances in machine learning, particularly deep learning, have enabled advancements in a variety of fields: recognition of faces, healthcare diagnosis, and processing of natural languages are a few examples (*Lakhanpal, Gupta & Agrawal, 2015*). However, they require an enormous amount of storage space and computing power and require the complex processing of large amounts of data in both central and decentralised data centers. Because the entire concept of moving processing to the edge is based on ultra-reliable low-latency communications (URLLC), distributed, trustworthy, and low-latency edge ML models built on local data. Edge ML offers low cost and minimal latency for mission-critical patient-facing IoT sensor devices. Any combination of the three basic fields of machine learning (supervised, unsupervised, and reinforcement learning) can be employed in a distributed healthcare system with an AI-integrated 5G infrastructure.

A recent survey provides more information about these strategies and how they relate to edge platforms (*Park et al., 2019*). AI can be deployed at the edge by partitioning helpers and devices. Each device generates a local learning model and transfers it to a helper that gathers all models from multiple devices (*Dean et al., 2012*; *Jin et al., 2016*). If one device runs out of memory, a local model can be partitioned and spread among multiple devices. In this case, the intermediate model is moved back and forth between the devices during

training. The application of AI and edge computing in 5G includes remote surgery, diagnosis, data and vital sign monitoring for patients. And the doctors can operate remotely (*Hassan, Yau & Wu, 2019*).

## Integration of blockchain technology for enhanced security and privacy

Blockchain technology creates an unchangeable record of transactions that is spread among participants in a ledger to guarantee secure and dispersed transactions. A block containing the transactions is connected to the chain. We clarify blockchain technology, which combines three existing technology such as consensus, distributed ledger, cryptography, and protocols (*Ismail, Materwala & Zeadally, 2019*).

### Distributed ledger

By doing away with the requirement for a reliable third party in digital relationships, the likelihood of a single point of failure is reduced by the distributed ledger (DL). Blockchain is a peer-to-peer network where every network node stores a synced copy of the ledger. In the event that a central administrative body holds the data, it is not possible to retrieve the original data from the other nodes in the event of a node failure or malicious behavior. As a result, blockchain enhances fault tolerance.

### Consensus protocols

Blockchain participants utilize consensus protocols (CPs) to reach a consensus on a single state for updating the ledger. The more nodes in the network that confirm a change in status, the higher the level of security in the network. Blockchain creates new blocks and appends them to the chain using a consensus protocol that verifies transactions.

### Cryptography technology

Every participant in the blockchain network receives a secure digital identity created for them using cryptography (C) technology, which also serves to validate transactions. To do this, the participant possess a set of public and private keys.

### Layers in blockchain architecture

Blockchain architecture is made up of three layers which includes distributed computing, platform, and infrastructure layers (*Ismail, Materwala & Zeadally, 2019*). The hardware elements required to run the blockchain are all included in the infrastructure layer. It consists of nodes, or the users of the network. The following tasks are all capable of being carried out by a node: 1) Start transactions, 2) verify transactions and blocks, 3) generate blocks, and 4) keep a copy of the ledger. The storage component that keeps the ledger of transaction records is also a part of the infrastructure layer. The network infrastructure required for communication within a blockchain or between various blockchains makes up the other half of the infrastructure layer.

The distributed computing layer of the blockchain architecture guarantees local data access, fault tolerance, immutability, privacy, authenticity, and transaction data security.

Replicated among dispersed nodes linked in a peer-to-peer network, the transaction record ledger provides immutability and fault tolerance. The feature of blockchain known as immutability prevents transaction records from being changed once they have been updated in the ledger. In order to come to a consensus on the sequence of network transactions, ledger updates and the creation of the subsequent block, the blockchain network employs a consensus algorithm. Furthermore, the distributed computing layer is in charge of data privacy *via* hashing and user authentication utilizing encryption.

The public Bitcoin network, where each user is linked to a set of private and public keys, is the most often used blockchain. For transaction authentication and validation, a user needs a pair of public and private keys in order to propose a transaction, have it validated, and have it included to the block. While the public key is accessible to all users on the network, a user's private key is only known to that user. The user uses a hashing function to first hash the transaction data before starting a new transaction. The user's private key is used to encrypt the hashed data before it is sent over the network with the transaction data. Every network validator verifies the transaction to make sure the proposing node is authentic, the transaction data is intact and the node is capable of completing the transaction. To accomplish this, each complete node uses the public key of the proposing node to decrypt the encrypted transaction data and get the hash value. The transaction data is then hashed by the full node to create a hash value, which it then compares to the hash value that has been decrypted. This guarantees that the transaction data is from an authentic user and was not tampered (*Ismail, Materwala & Zeadally, 2019*).

## Integration of IoT with unified communications as a service

Unified communications as a service (UCaaS) is a cloud-based delivery paradigm that provides a wide range of communication and collaboration applications and services. This is an enterprise communication system in which all communications are streamlined *via* cloud delivery, allowing the industry to be more flexible in terms of human-financial resources and spending. Virtual apps (MS Teams, Zoom, Skype, YouTube, and WebEx) and mobile system apps (video, audio, SMS, and chat) are examples of UCaaS (*Mah, Skalna & Muzam, 2022*). Devices can be integrated with IoT sensors to transmit alerts or messages directly to communication channels, automate conversations, or initiate video on communication conferences based on sensor data or events using middleware to translate IoT protocols into unified communication systems. Integration of IoT and cloud-based technologies such as UCaaS has brought great transformation (*Mehmood et al., 2024*). Communication between remote or hybrid environments are made possible with UCaaS on a single platform. Scalability can be increased with the help of UCaaS which will cater to growing business needs. UCaaS solution enables an organization to integrate multiple devices such as laptops, desktops, and mobile phones into their solution. Following the development of UCaaS, many organisations with good hardware sought cheaper, faster, and more convenient alternatives to system integration rather than constructing their own (*Son et al., 2019*).

Some of the advantages of using UCaaS are reduction in cost of purchasing hardware for communication purposes, more scalability in communication, speed of implementation,

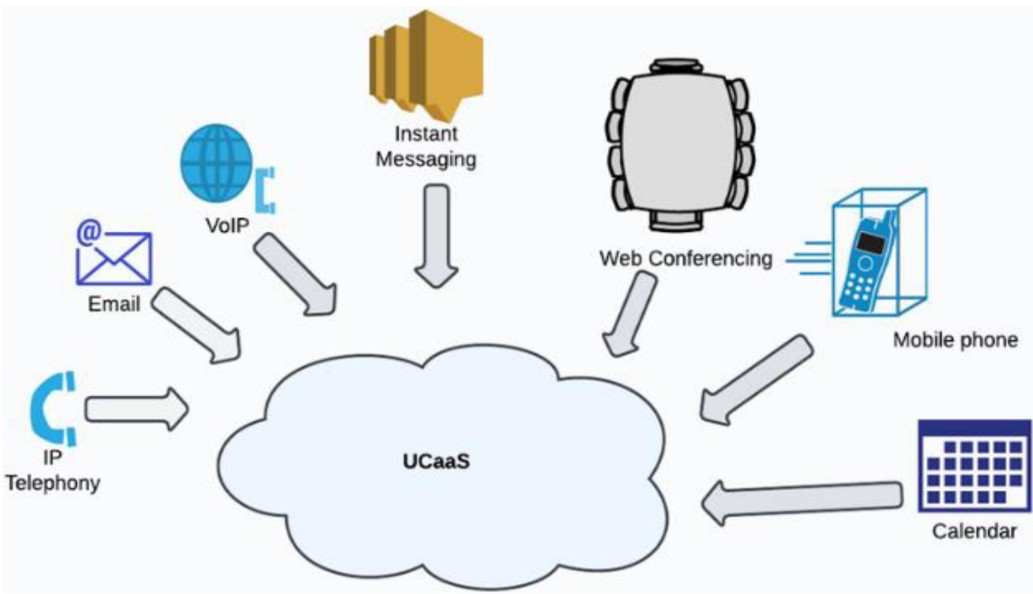

**Figure 18 Range of solutions available with UCaaS systems.**

availability of services regardless of geographical locations, improved communication and employee productivity, increased technical support by the service providers, unified management of communication systems, and low demand of staff. Though there are many advantages few challenges still exist. Some of them are –

*Security-related barriers* which includes loss of control over valuable data, unauthorized disclosure of data to competitors, UCaaS non-compliance solutions based on the security policies that have been implemented.

*Technical barriers* which includes lack of internet connection, network failure, problem in data migration, *etc*.

*Legal barriers* which includes problem in negotiating legal terms with the service providers, risk for non-compliance with the guidelines of the service providers.

*Psychological barriers* which includes reluctance of the employees to implement new communication and collaboration methods due to a lack of confidence, being accustomed to the present communication and collaboration systems, lack of information, and understanding about the importance of UCaaS. Range of solutions available with UCaaS has been depicted in Fig. 18.

## CONCLUSION

Many technical solutions are being developed by researchers worldwide. These technological solutions will help improve smart healthcare facilities by leveraging the mobilisation potential of the IoHT and addressing the complexity of healthcare services currently offered. A lot of research and development goes into making IoHT services and apps work more smoothly. With minimal processing latency, edge and fog computing is a fascinating area of future cellular networks designed to accommodate a variety of IoT devices. Our main focus in our survey article was its use in IoHT, despite the wide range of

use cases. This article provides an overview of current fog and edge-based healthcare models that support IoHT infrastructure and facilitate the transmission and response of intelligent healthcare data. This article also provides a brief overview of IoHT and lists the requirements for integrating edge and fog computing technologies into IoHT. Also it analyses a variety of machine learning algorithms and techniques for efficient resource management on the Internet of Things. The review explains how these machine learning techniques are applied in different scenarios. This overview study provides in-depth insights into enabling technologies and industry trends by exhibiting past, present, and future trends in conjunction with multiple current improvements in the IoHT area. This review also gives an overview of the advancement of IoHT due to the contribution of various technologies. It will drive the industry's development of user-centric, cost-effective healthcare devices and build a network of intelligent, sensor-based, connected devices by opening new services in the near future. Furthermore, this work explores possible directions for future research by addressing the issues and difficulties associated with IoHT-based solutions.

## LIST OF IMPORTANT ABBREVIATIONS

**IoT**      Internet of Things
**IoHT**     Internet of Health Things
**AI**       Artificial Intelligence
**ML**       Machine Learning
**DL**       Deep Learning
**UCaaS**    Unified Communication as a Service
**NB-IoT**   Narrow Band IoT
**EMR**      Electronic Medical Record
**ARM**      Advanced RISC Machines
**AMD**      Advanced Micro Device
**SLA**      Service Level Agreements
**HIPAA**    Health Insurance Portability and Accountability Act
**DES**      Data Encryption Standard
**AES**      Advanced Encryption Standard
**WBANs**    Wireless Body Area Networks
**IPS**      Intrusion Prevention System
**IoMT**     Internet of Medical Things
**IoV**      Internet of Vehicles
**DFDM**     Digital Forensic Data Model
**IDFM**     Integrated Digital Forensic Process Model
**MEC**      Multi-Access Edge Computing
**QoS**      Quality of Service
**EHR**      Electronic Health Records
**SHM**      Smart Health Monitoring
**NFV**      Network Function Virtualization

| **SDN** | Software-Defined Networking |
|---|---|
| **MIMO** | Multiple-input Multiple-output |
| **RMAC** | Random MAC |
| **AKA** | Authentication and Key Agreement |
| **VM** | Virtual Machine |
| **CPU** | Central Processing Unit |
| **DC** | Data Center |
| **URLLC** | Ultra-Reliable Low-Latency Communications |
| **DL** | Distributed Ledger |
| **CPs** | Consensus Protocols |

### Funding
This work was supported by Vellore Institute of Technology, Chennai. The funders had no role in study design, data collection and analysis, decision to publish, or preparation of the manuscript.

### Grant Disclosures
The following grant information was disclosed by the authors:
Vellore Institute of Technology, Chennai.

### Competing Interests
The authors declare that they have no competing interests.

### Author Contributions
- Deepika Rajagopal conceived and designed the experiments, performed the experiments, analyzed the data, prepared figures and/or tables, authored or reviewed drafts of the article, and approved the final draft.
- Pradeep Kumar Thimma Subramanian conceived and designed the experiments, performed the experiments, analyzed the data, prepared figures and/or tables, authored or reviewed drafts of the article, and approved the final draft.

### Data Availability
This is a literature review.

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
