# Peer review of "AI augmented edge and fog computing for Internet of Health Things (IoHT)"

_PeerJ Computer Science, doi:10.7717/peerj-cs.2431_

## Round 0.1 · original submission · Major Revisions

Both the reviewers agree on requesting major revisions. Considering the importance of this field I would ask to address all their comments, in order to provide readers an improved review paper.

Reviewer 1 ·

Basic reporting

The manuscript presents a well-structured and professionally written piece that contributes to the field of IoHT. The author has made commendable efforts to maintain clarity and unambiguity throughout the text, using professional English effectively. The article is of broad interest and aligns with the interdisciplinary scope of the journal.

Experimental design

The manuscript provides a comprehensive overview of the Internet of Health Things (IoHT) technology, demonstrating a good grasp of the subject matter. However, it lacks depth and original insights in some key areas, which could be addressed by delving into more detailed analysis and offering the author's unique perspectives on current challenges and future directions.

Furthermore, the literature cited is not sufficiently representative of the field, potentially limiting the review's comprehensiveness. It is recommended that the author includes a broader range of seminal and recent works to strengthen the credibility and relevance of the review.

Lastly, while the review is logically organized, there is room for improvement in the flow and coherence of the arguments presented. Enhancing the logical progression of the sections and ensuring a smooth transition between topics will greatly benefit the manuscript's clarity and impact.

Validity of the findings

This review article presents a solid foundation in discussing the current state of the field. However, the discussion and conclusion sections would benefit from a broader perspective that encompasses a wider range of topics. It is crucial to address the multifaceted nature of the subject matter to provide a more comprehensive analysis.

Additionally, the manuscript would be significantly enhanced by clearly articulating its unique contributions. The author should emphasize the distinctive aspects of this review compared to others in the field and highlight its value proposition. A clear demonstration of how this review advances the discourse, either through novel insights or a fresh synthesis of existing knowledge, is essential.

To achieve this, the author may consider reorganizing the discussion to reflect on the broader implications of the findings and to connect the conclusions more explicitly with the contributions of the review. This will not only strengthen the manuscript's impact but also solidify its position as a valuable addition to the literature.

Additional comments

1. The figures included in the manuscript require adjustment in format and clarity. It is essential that all labels and numerical values are legible to ensure readers can fully understand the visual data presented.
2. To enhance clarity, especially for readers who may be less familiar with the field, the author should consider adding a list of abbreviations for domain-specific terminology at the beginning of the article.
3. Each figure, table, and equation should be accompanied by more detailed explanations to strengthen the logical connection between the visual elements and the textual content.
4. The current number of references is insufficient for a review paper. The author should expand the literature review to include a broader and more representative selection of sources, ensuring both quantity and quality.
5. The manuscript would benefit from an in-depth discussion of state-of-the-art technologies in AI augmented IoHT. Particularly, the author should explore the use of advanced vision transformers, such as the 'RanMerFormer: Randomized vision transformer with token merging for brain tumor classification,' to demonstrate the cutting-edge applications of IoHT. This inclusion would enhance the review's relevance and showcase the potential of these technologies in the field.
6. The section on the Evolutionary Algorithm (EA) is somewhat underdeveloped. The author is encouraged to expand this section and reference the work 'An evolutionary attention-based network for medical image classification' for a more comprehensive treatment.
7. The discussion on the Particle Swarm Optimization (PSO) algorithm could be better connected to current research within the IoHT domain. The author should consider referencing 'Alcoholism detection via GLCM and particle swarm optimization' and expanding on the PSO's role in this context.
8. The author should explore the broader implications of the technologies discussed, considering their impact on various aspects of IoHT and how they contribute to the advancement of the field.
9. The manuscript requires a thorough review for grammar and language fluency. It is important to correct any grammatical errors and improve sentence structure to ensure the text is coherent and professional.

Cite this review as

Reviewer 2 ·

Basic reporting

The paper surveys the literature about IoHT and the use of edge, fog computing with cloud computing to meet the requirements of this domain, such as reducing delay, reliability, privacy and security. In general the paper offers good information. However the structure of the abstract and introduction needs revision. Why such review is necessary -> this should be clear in the abstract and introduction. The first paragraph in the introduction presents the development of IT in healthcare with too many details. This paragraph can be divided into two.

Experimental design

The study design is appropriate. some minor issues need revision
- write the full terms of the abbreviation where it appears first time in the text, e.g. ML in the introduction (not abstract)
- minor English language check
- is it IoHT or IOHT?
- the figures should explain themselves, adding more words to figure captions is necessary ?
- What is the next step after this survey? provide future work recommendation with more details on specific topics, e.g what need to be done in security, reliability, delay reduction, privacy
- The blockchain domain needs a section in this review it is highly used and integrated with healthcare

Validity of the findings

The findings are valid but some important references in this domain are missing. The author can look at the following references for more communication protocols, security, privacy, and delay.
- Edge–fog–cloud computing hierarchy for improving performance and security of NB-IoT-based health monitoring systems
-Blockchain framework for Enhancing NB-IoT security and authentication: Health monitoring system as a case

Cite this review as

---

## Round 0.2 · accepted · Accept

The reviewers agree that you have addressed their comments, therefore the article is ready for publication.

Reviewer 1 ·

Basic reporting

The authors have addressed all my queries, and I believe the manuscript is now suitable for publication in the journal. I recommend acceptance.

Experimental design

No comment

Validity of the findings

No comment

Additional comments

No comment

Cite this review as

Reviewer 2 ·

Basic reporting

The authors made the required modifications

Experimental design

The authors made the required modifications

Validity of the findings

The authors made the required modifications

Additional comments

The authors made the required modifications

Cite this review as